# Behavior-Regularized Diffusion Policy Optimization for Offline Reinforcement Learning

**Chen-Xiao Gao** [1]  **Chenyang Wu** [1]  **Mingjun Cao** [1]  **Chenjun Xiao** [2]  **Yang Yu** [1]  **Zongzhang Zhang** [1]

## Abstract

Behavior regularization, which constrains the policy to stay close to some behavior policy, is widely used in offline reinforcement learning (RL) to manage the risk of hazardous exploitation of unseen actions. Nevertheless, existing literature on behavior-regularized RL primarily focuses on explicit policy parameterizations, such as Gaussian policies. Consequently, it remains unclear how to extend this framework to more advanced policy parameterizations, such as diffusion models. In this paper, we introduce BDPO, a principled behavior-regularized RL framework tailored for diffusion-based policies, thereby combining the expressive power of diffusion policies and the robustness provided by regularization. The key ingredient of our method is to calculate the Kullback-Leibler (KL) regularization analytically as the accumulated discrepancies in reverse-time transition kernels along the diffusion trajectory. By integrating the regularization, we develop an efficient two-time-scale actor-critic RL algorithm that produces the optimal policy while respecting the behavior constraint. Comprehensive evaluations conducted on synthetic 2D tasks and continuous control tasks from the D4RL benchmark validate its effectiveness and superior performance. The code and experiment results of BDPO are available on the project webpage.

## 1. Introduction

Despite its huge success in industrial applications such as robotics (Kumar et al., 2021), game AI (Vinyals et al., 2019),

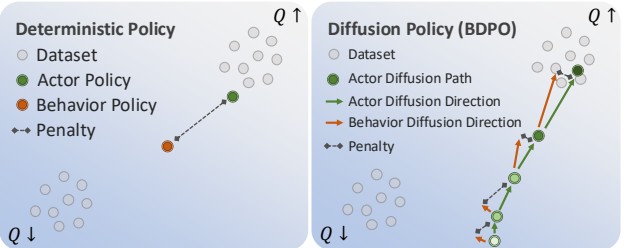

*Figure 1.* Illustration of the behavior-regularized RL framework with different policy parameterizations. Unimodal policies, such as deterministic policies (left), compute the behavior as the center of mass and therefore lead to misleading regularizations; while our method (right) harnesses the flexibility of diffusion models, and the regularization is calculated as the accumulated discrepancies in diffusion directions of the actor and the behavior diffusion.

and generative model fine-tuning (Ouyang et al., 2022), reinforcement learning (RL) algorithms typically require millions of online interactions with the environment to achieve meaningful optimization (Haarnoja et al., 2018; Schulman et al., 2017). Given that online interaction can be hazardous and costly in certain scenarios, offline RL, which focuses on optimizing policies with static datasets, emerges as a practical and promising avenue (Levine et al., 2020).

However, without access to the real environment, RL algorithms tend to yield unrealistic value estimations for actions that are absent from the dataset (Levine et al., 2020; Fujimoto & Gu, 2021). Since RL necessitates querying the values of unseen actions to further refine its policy beyond the dataset, it is prone to exploiting the values of those unseen actions, leading to serious overestimation in value function. To manage this risk, one predominant approach is to employ behavior regularization (Fujimoto et al., 2019; Kumar et al., 2019; Wu et al., 2019; Tarasov et al., 2023), which augments the usual RL objectives by incorporating penalty terms that constrain the policy to remain close to the behavior policy that collects the datasets. In this way, the policy is penalized for choosing unreliable out-of-distribution (OOD) actions and thus exercises the principle of pessimism in the face of uncertainty.

Most preliminary works in offline RL assume explicit policy distributions (Fujimoto & Gu, 2021; Haarnoja et al.,

---

[1]National Key Laboratory for Novel Software Technology, Nanjing University, China & School of Artificial Intelligence, Nanjing University, China [2]The Chinese University of Hong Kong, Shenzhen, China. Correspondence to: Zongzhang Zhang <zzzhang@nju.edu.cn>.

*Proceedings of the 42nd International Conference on Machine Learning*, Vancouver, Canada. PMLR 267, 2025. Copyright 2025 by the author(s).

2018; Christodoulou, 2019); for instance, the agent's policy is often modeled as a Gaussian distribution with parameterized mean and diagonal covariance, or a deterministic distribution that directly outputs the action. Despite the computational efficiency, such assumptions present significant challenges within the aforementioned behavior-regularized framework (Wang et al., 2023; Hansen-Estruch et al., 2023; Chen et al., 2023). The underlying difficulty originates from the multi-modality of the distributions and manifests in two ways: 1) the dataset may be collected from multiple policy checkpoints, meaning that using unimodal distributions to approximate the behavior policy for regularization can introduce substantial approximation errors (see Figure 1); 2) the optimal policy distribution in behavior-regularized RL framework is actually the Boltzmann distribution of the optimal value function (Haarnoja et al., 2017; Nair et al., 2020), which is inherently multi-modal as well. Such limitation motivates researchers to explore diffusion models, which generate actions through a series of denoising steps, as a viable parameterization of the policy (Wang et al., 2023; Hansen-Estruch et al., 2023; Chen et al., 2024a;b; Fang et al., 2024; Zhang et al., 2024b).

Nonetheless, extending the behavior-regularized RL framework to diffusion policies faces specific challenges. Although diffusion models are capable of generating high-fidelity actions, the log-probability of the generated actions is difficult to compute (Song et al., 2021). Consequently, the calculation of regularization terms, such as Kullback-Leibler (KL) divergence, within the behavior-regularized RL framework remains ambiguous (Zhang et al., 2024b; Wang et al., 2024). Besides, it is also unclear how to improve diffusion policies while simultaneously imposing effective regularization. In this paper, we introduce BDPO, which provides an efficient and effective framework tailored for diffusion policies. Specifically:

1) Framing the reverse process of diffusion models as an MDP, we propose to implement the KL divergence w.r.t. the diffusion generation path, rather than the clean action samples (Figure 1);

2) Building upon this foundation, we propose a two-timescale actor-critic method to optimize diffusion policies. Instead of differentiating the policy along the entire diffusion path, BDPO estimates the values at intermediate diffusion steps to amortize the optimization, offering efficient computation, convergence guarantee, and state-of-the-art performance;

3) Experiments conducted on synthetic 2D datasets reveal that our method effectively approximates the target distribution. Furthermore, when applied to continuous control tasks provided by D4RL, BDPO demonstrates superior performance compared to baseline offline RL algorithms.

## 2. Related Work

**Offline RL.** To improve beyond the interaction experience, RL algorithms need to query the estimated values of unseen actions for optimization. In offline scenarios, such distribution shift tends to cause serious overestimation in value functions due to the lack of *corrective feedback* (Kumar et al., 2020a). To mitigate this, algorithms like CQL (Kumar et al., 2020b), EDAC (An et al., 2021), and PBRL (Bai et al., 2022) focus on penalizing the Q-values of OOD actions to prevent the over-estimation issue. Behavior regularization provides another principled way to mitigate the distribution shift problem, by adding penalties for deviation from the dataset policy during the stage of policy evaluation, policy improvement (Fujimoto et al., 2019; Fujimoto & Gu, 2021; Ran et al., 2023), or sometimes both (Wu et al., 2019; Tarasov et al., 2023). Another line of research, also based on the behavior-regularized RL framework, approaches offline RL by performing in-sample value iteration (Kostrikov et al., 2022; Garg et al., 2023; Xu et al., 2023), thus eliminating OOD queries from the policy and directly approximating the optimal value function. Lastly, model-based offline RL methods (Yu et al., 2020; Jia et al., 2024; Sun et al., 2023; Luo et al., 2024) introduce learned dynamics models that generate synthetic experiences to alleviate data limitations, providing extra generalization compared to model-free algorithms.

**Diffusion Policies in Offline RL.** There has been a notable trend towards the applications of expressive generative models for policy parameterization (Janner et al., 2022), dynamics modeling (Micheli et al., 2023; Alonso et al., 2024), trajectory planning (Chen et al., 2021; Ajay et al., 2023; Gao et al., 2024), and representation learning (Shribak et al., 2024). In offline RL, several works employ diffusion models to approximate the behavior policy used for dataset collection. To further improve the policy, they utilize in-sample value iteration to derive the Q-values, and subsequently select action candidates from the behavior diffusion model (Hansen-Estruch et al., 2023; Chen et al., 2023) or use the gradient of Q-value functions to guide the generation (Lu et al., 2023; Mao et al., 2024). However, the performance of these methods is limited, as the actions come from behavior diffusion. Alternatively, SRPO (Chen et al., 2024a) and DTQL (Chen et al., 2024b) maintain a simple one-step policy for optimization while harnessing the diffusion model or diffusion loss to implement behavior regularization. This yields improved computational efficiency and performance in practice; however, the single-step policy still restricts expressiveness and fails to encompass all of the modes of the optimal policy. A wide range of works therefore explore using diffusion models as the actor. Among them, DAC (Fang et al., 2024) formulates the optimization as a noise-regression problem and proposes to align the output from the policy with the gradient of the Q-value func-

tions. Diffusion-QL (Wang et al., 2023) optimizes the actor by back-propagating the gradient of Q-values throughout the entire diffusion path. This results in a significant memory footprint and computational overhead, and EDP (Kang et al., 2023) proposes to use action approximations to alleviate the cost. In contrast, `BDPO` maintains value functions for intermediate diffusion steps, thereby amortizing the optimization cost while also keeping the optimization precise.

## 3. Preliminaries

**Behavior-Regularized Offline RL.** We formalize the task as a Markov Decision Process (MDP) $\langle \mathcal{S}, \mathcal{A}, T, R, \gamma \rangle$, where $\mathcal{S}$ is the state space, $\mathcal{A}$ is the action space, $T(s'|s,a)$ denotes the transition function, $R(s,a)$ is a bounded reward function, and $\gamma$ is the discount factor. In offline RL, an agent is expected to learn a policy $\pi : \mathcal{S} \to \Delta(\mathcal{A})$ to maximize the expected discounted return $\mathbb{E}_{\pi}[\sum_{t=0}^{\infty} \gamma^t R(s_t, a_t)]$ with an offline dataset $\mathcal{D} = \{(s_t, a_t, s_{t+1}, r_t)\}$, where $s_t, s_{t+1} \in \mathcal{S}$, $a_t \in \mathcal{A}$, and $r_t = R(s_t, a_t) \in \mathbb{R}$. We consider the behavior-regularized RL objective, which augments the original RL objective by regularizing the policy towards some behavior policy $\nu$:

$$\max_{\pi} \ \mathbb{E}_{\pi}\left[\sum_{t=0}^{\infty} \gamma^t (r_t - \eta D_{\mathrm{KL}}\left[\pi(\cdot|s_t) \| \nu(\cdot|s_t)\right])\right], \quad (1)$$

where $D_{\mathrm{KL}}$ is the Kullback-Leibler (KL) divergence and $\eta > 0$ controls the regularization strength. In offline RL, Eq. (1) is widely employed by setting $\nu$ as the policy $\pi_{\mathcal{D}}$ that collects the dataset to prevent the exploitation of the out-of-dataset actions. Besides, when setting $\nu$ as the uniform distribution, Eq. (1) equates to the maximum-entropy RL in online scenarios up to some constant.

To solve Eq. (1), a well-established method is soft policy iteration (Haarnoja et al., 2018; Wu et al., 2019). Specifically, we define the soft value functions as

$$V^{\pi}(s) = \mathbb{E}_{\pi}\left[\sum_{t=0}^{\infty} \gamma^t \left(r_t - \eta \log \frac{\pi(a_t|s_t)}{\nu(a_t|s_t)}\right)\right], \quad (2)$$

where the expectation is taken w.r.t. random trajectories generated by $\pi$ under the initial condition $s_0 = s$ and $a_0 = a$. The soft $Q$-value function in this framework can be solved by the repeated application of the soft Bellman operator $\mathcal{B}^{\pi}$:

$$\mathcal{B}^{\pi} Q^{\pi}(s,a) = R(s,a) + \gamma \mathbb{E}\left[Q^{\pi}(s',a') - \eta \log \frac{\pi(a'|s')}{\nu(a'|s')}\right], \quad (3)$$

where $s' \sim T(\cdot|s,a)$ and $a' \sim \pi(\cdot|s')$. For policy improvement, we can update the policy using the objective:

$$\max_{\pi} \ \mathbb{E}_{a \sim \pi(\cdot|s)}\left[Q^{\pi}(s,a)\right] - \eta D_{\mathrm{KL}}\left[\pi(\cdot|s) \| \nu(\cdot|s)\right]. \quad (4)$$

Note that regularization is added for both $Q$-value functions and the policy. By iterating between policy evaluation and

improvement, the performance of the policy defined by Eq. (1) is guaranteed to improve (Haarnoja et al., 2018).

**Diffusion Models.** Diffusion models (Ho et al., 2020; Song et al., 2021) consist of a forward Markov process which progressively perturbs the data $x^0 \sim q_0$ to data that approximately follows the standard Gaussian distribution $x^N \sim q_N$, and a reverse Markov process that gradually recovers the original $x^0$ from the noisy sample $x^N$. The transition of the forward process $q_{n+1|n}$ usually follows Gaussian distributions:

$$q_{n|n-1}(x^n|x^{n-1}) = \mathcal{N}(x^n; \sqrt{1-\beta_n}x^{n-1}, \beta_n I), \quad (5)$$

where $\{\beta_n\}_{n=1}^N$ is specified according to the *noise schedule* and $I$ denotes the Identity matrix. Due to the closure property of Gaussian distributions, the marginal distribution of $x^n$ given $x^0$ can be specified as:

$$q_{n|0}(x^n|x^0) = \mathcal{N}(x^n; \sqrt{\bar{\alpha}_n}x^0, (1-\bar{\alpha}_n)I), \quad (6)$$

where $\alpha_n = 1 - \beta_n, \bar{\alpha}_n = \prod_{n'=1}^n \alpha_{n'}$. The transition of the reverse process can be derived from Bayes' rule,

$$q_{n-1|n}(x^{n-1}|x^n) = \frac{q_{n|n-1}(x^n|x^{n-1})q_{n-1}(x^{n-1})}{q_n(x^n)}. \quad (7)$$

However, it is usually intractable, and therefore we use a parameterized neural network $p_{n-1|n}^{\theta}$ to approximate the reverse transition, which is also a Gaussian distribution with parameterized mean:

$$p_N^{\theta}(x^N) = \mathcal{N}(0, I),$$
$$p_{n-1|n}^{\theta}(x^{n-1}|x^n) = \mathcal{N}(x^{n-1}; \mu_n^{\theta}(x^n), \sigma_n^2 I), \quad (8)$$

where $\sigma_n = \sqrt{\frac{1-\bar{\alpha}_{n-1}}{1-\bar{\alpha}_n}\beta_n} \approx \sqrt{\beta_n}$. The learning objective is matching $p_{n-1|n}^{\theta}(x^{n-1}|x^n)$ with the posterior $q_{n-1|n,0}(x^{n-1}|x^n, x^0)$:

$$\mathcal{L}_{\mathrm{diff}}(\theta) =$$
$$\mathbb{E}_{n,x^0,x^n}\left[D_{\mathrm{KL}}\left[q_{n-1|n,0}(x^{n-1}|x^n, x^0) \| p_{n-1|n}^{\theta}(x^{n-1}|x^n)\right]\right], \quad (9)$$

where $x^0 \sim q_0, x^n \sim q_{n|0}(\cdot|x^0)$. Upon the completion of training, it can be shown that the objective Eq. (9) yields

$$p_{n-1|n}^{\theta}(x^{n-1}|x^n) \approx q_{n-1|n}(x^{n-1}|x^n). \quad (10)$$

Simplified training objectives can be derived by reparameterizing $\mu^{\theta}$ with noise prediction or score networks (Ho et al., 2020; Song et al., 2021). After training, the generation process begins by sampling $\hat{x}^N \sim \mathcal{N}(0, I)$, followed by iteratively applying $p_{n-1|n}^{\theta}$ to generate the final samples $\hat{x}_0$, which approximately follow the target distribution $q_0$.

**Diffusion Policy.** Diffusion policies are conditional diffusion models that generate action $a$ on a given state $s$.

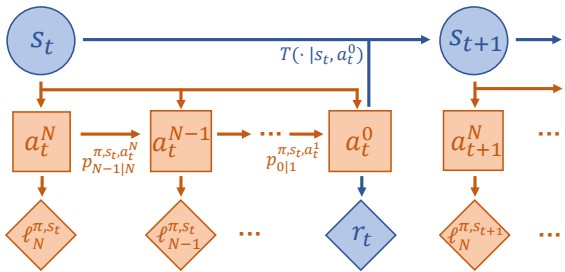

*Figure 2.* Semantic illustration of the interplay between diffusion policies and the environment. We use orange to denote the transition $p_{n-1|n}^{\pi,s_t,a^n}$ and the penalty $\ell_n^{\pi,s_t}$ (see Section 4.1) associated with the diffusion generation process, whereas blue signifies the transition $T(\cdot|s_t, a_t^0)$ and the reward $r_t$ from the original environment MDP.

In this paper, we will use $p^\pi$ and $p^{\pi,s}$ to denote the diffusion policy and its state-conditioned version, respectively. Similarly, we will use $p_{n-1|n}^{\pi,s,a^n}$ as a shorthand for the single-step reverse transition conditioned on $a^n$, i.e., $p_{n-1|n}^{\pi,s,a^n} = p_{n-1|n}^{\pi,s}(\cdot|a_n)$. At timestep $t$ of the environment MDP, the agent observes the state $s_t$, drives the reverse diffusion process $a^{0:N} \sim p_{0:N}^{\pi,s_t}$ as defined in Eq. (8), and takes $a^0$ as its action $a_t$. For action $a_t^n$, we will use $t$ in the subscript to denote the timestep in the environment MDP, while using $n$ in the superscript to denote the diffusion steps. A semantic illustration of the environment MDP and the reverse diffusion is provided in Figure 2.

## 4. Method

### 4.1. Pathwise KL Regularization

In behavior-regularized RL, previous methods typically regularize action distribution, i.e., the marginalized distribution $p_0^{\pi,s}$. Instead, we shift our attention to the KL divergence with respect to the diffusion path $a^{0:N}$, which can be further decomposed thanks to the Markov property:

$$
\begin{aligned}
& D_{\mathrm{KL}}\left[p_{0:N}^{\pi,s} \| p_{0:N}^{\nu,s}\right] \\
&= \mathbb{E}\left[\log \frac{p_{0:N}^{\pi,s}(a^{0:N})}{p_{0:N}^{\nu,s}(a^{0:N})}\right] \\
&= \mathbb{E}\left[\log \frac{p_N^{\pi,s}(a^N) \prod_{n=1}^N p_{n-1|n}^{\pi,s}(a^{n-1}|a^n)}{p_N^{\nu,s}(a^N) \prod_{n=1}^N p_{n-1|n}^{\nu,s}(a^{n-1}|a^n)}\right] \\
&= \mathbb{E}\left[\log \frac{p_N^{\pi,s}(a^N)}{p_N^{\nu,s}(a^N)} + \sum_{n=1}^N \log \frac{p_{n-1|n}^{\pi,s,a^n}(a^{n-1})}{p_{n-1|n}^{\nu,s,a^n}(a^{n-1})}\right] \\
&= \mathbb{E}\left[\sum_{n=1}^N D_{\mathrm{KL}}\left[p_{n-1|n}^{\pi,s,a^n} \| p_{n-1|n}^{\nu,s,a^n}\right]\right].
\end{aligned}
\tag{11}
$$

Here, the expectation is taken w.r.t. $a^{0:N} \sim p_{0:N}^{\pi,s}$, $p^\nu$ is the behavior diffusion policy trained via Eq. (9) on the offline dataset $\mathcal{D}$ to approximate the data-collection policy, and

the last equation holds due to $p_N^{\pi,s} = p_N^{\nu,s} = \mathcal{N}(0, I)$. In the following, we abbreviate $D_{\mathrm{KL}}\left[p_{n-1|n}^{\pi,s,a^n} \| p_{n-1|n}^{\nu,s,a^n}\right]$ with $\ell_n^{\pi,s}(a^n)$.

Based on this decomposition, we present the pathwise KL-regularized RL problem.

**Definition 4.1.** *(Pathwise KL-Regularized RL)* Let $p^\nu$ be the behavior diffusion process. The pathwise KL-regularized RL problem seeks to maximize the following objective

$$
\max_{p^\pi} \mathbb{E}\left[\sum_{t=0}^\infty \gamma^t \left(R(s_t, a_t^0) - \eta \sum_{n=1}^N \ell_n^{\pi,s_t}(a_t^n)\right)\right]. \tag{12}
$$

where the expectation is taken w.r.t. trajectories following $p^\pi$ and the dynamics $T$.

Although our problem calculates the accumulated discrepancies along the diffusion path as penalties, it actually preserves the same optimal solution to Eq. (1), which only regularizes the action distribution at diffusion step $n = 0$. The following theorem captures the equivalence.

**Theorem 4.2.** *(Proof in Appendix C.2) Let $p^\nu$ be the behavior diffusion process. The optimal diffusion policy $p^*$ of the pathwise KL-regularized RL problem in Eq. (12) is also the optimal policy $\pi^*$ of the KL regularized objective in Eq. (1), in the sense that $\pi^*(a|s) = \int p_{0:N}^{*,s}(a^{0:N})\delta(a - a^0)\mathrm{d}a^{0:N}$ $\forall s \in \mathcal{S}$, where $\delta$ is the Dirac delta function.*

That said, we can safely optimize Eq. (12) and ultimately arrive at the solution that also maximizes the original KL-regularized RL problem. As will be demonstrated in the following sections, our formulation of the penalty leads to an analytical and efficient algorithm, since each single-step reverse transition is tractable.

### 4.2. Actor-Critic across Two Time Scales

To solve the pathwise KL-regularized RL (Eq. (12)), we can employ an actor-critic framework. Specifically, we maintain a diffusion policy $p^\pi$ and a critic $Q^\pi$. The update target of the critic $Q^\pi$ is specified as the standard TD target in the environment MDP:

$$
\mathcal{B}^\pi Q^\pi(s, a) = R(s, a) + \gamma \mathbb{E}\left[Q^\pi(s', a'^0) - \eta \sum_{n=1}^N \ell_n^{\pi,s'}(a'^n)\right], \tag{13}
$$

i.e., we sample diffusion paths $a'^{0:N}$ at the next state $s'$, calculate the Q-values $Q^\pi(s', a'^0)$ and the accumulated penalties along the path, and perform a one-step TD backup. For the diffusion policy, its objective can be expressed as:

$$
\max_{p^{\pi,s}} \mathbb{E}_{a^{0:N} \sim p_{0:N}^{\pi,s}}\left[Q^\pi(s, a^0) - \eta \sum_{n=1}^N \ell_n^{\pi,s}(a^n)\right]. \tag{14}
$$

*Figure 3.* Semantic illustration of the TD backup for the Q-value function $Q^\pi$ (blue) and diffusion value function $V_n^{\pi,s}$ (orange). The update of $Q^\pi$ (Eq. (13)) requires reward, penalties along the diffusion trajectory, and the Q-values at the next state. The update of $V_n^{\pi,s}$ (Eq. (15)) involves the single-step penalty and the diffusion value at the next diffusion step $n-1$.

At first glance, one might treat the pathwise KL as a whole and optimize the diffusion policy by back-propagating the gradient throughout the sampled path $a^{0:N}$, similar to Diffusion-QL (Wang et al., 2023). Nevertheless, this requires preserving the computation graph of all diffusion steps and therefore incurs considerable computational overhead. Moreover, the inherent stochasticity of the diffusion path introduces significant variance during optimization, which can hinder the overall performance.

Instead, we opt for a bi-level TD learning framework. The upper level updates $Q^\pi$ according to Eq. (13) and operates in the environment MDP. In the lower level, we maintain *diffusion value functions* $V_n^{\pi,s}(a^n)$, which are designed to estimate the values at intermediate diffusion steps. The TD target of $V_n^{\pi,s}$ is specified as:

$$\mathcal{T}_0^\pi V_0^{\pi,s}(a^0) = Q^\pi(s, a^0),$$
$$\mathcal{T}_n^\pi V_n^{\pi,s}(a^n) = -\eta \ell_n^{\pi,s}(a^n) + \underset{p_{n-1|n}^{\pi,s,a^n}}{\mathbb{E}}\left[V_{n-1}^{\pi,s}(a^{n-1})\right].$$
$$(15)$$

Intuitively, $V_n^{\pi,s}(a^n)$ receives the state $s$, intermediate generation result $a^n$, and the diffusion step $n$ as input, and estimates the expected cumulative penalties of continuing the diffusion path at $a^n$ and step $n$. It operates completely inside each environment timestep and performs TD updates across the diffusion steps. The semantic illustration of the value function backups is presented in Figure 3. The following proposition demonstrates the convergence of policy evaluation.

With the help of $V_n^{\pi,s}$, we can update the policy according to

$$\max_{p_{n-1|n}^{\pi,s,a^n}} \; -\eta \ell_n^{\pi,s}(a^n) + \underset{p_{n-1|n}^{\pi,s,a^n}}{\mathbb{E}}\left[V_{n-1}^{\pi,s}(a^{n-1})\right].$$
$$(16)$$

Note that the maximization is over every state $s$, diffusion step $n$, and intermediate action $a^n$. In this way, we only require single-step reverse diffusion during policy improvement, providing an efficient solution to the problem defined in Eq. (16).

Through simple recursion, we can demonstrate that updating $p^\pi$ with Eq. (16) leads to the same optimal solution to the original policy objective Eq. (14).

**Algorithm 1** Behavior-Regularized Diffuion Policy Optimization (BDPO)

---

**Input**: Offline dataset $\mathcal{D}$, initialized diffusion policy $p^\pi$ and $p^\nu$, Q-value networks $Q_{\psi_k}$, diffusion value networks $V_{\phi_k}$.

1: // Pretrain $p^\nu$
2: Pretrain the behavior diffusion policy $p^\nu$ via Eq. (9)
3: Initialize $p^\pi$ with $p^\nu$
4: // Train $p^\pi$
5: **for** $i = 1, 2, \cdots, N_{\text{total\_steps}}$ **do**
6:      Sample minibatch $B = \{(s_i, a_i, s_i', r_i)\}_{i=1}^{|B|}$ from $\mathcal{D}$
7:      Update $\{Q_{\psi_k}\}_{k=1}^K$ via Eq. (19) using $B$
8:      Extend each $(s_i, a_i, s_i', r_i)$ in $B$ by sampling $n \sim U[1, N]$ and $a_t^n \sim q_{n|0}(\cdot|a_t)$
9:      Update $\{V_{\phi_k}\}_{k=1}^K$ via Eq. (19) using $B$
10:      **if** $i$ % actor_update_interval == 0 **then**
11:          Update $p^\pi$ by performing gradient ascend with the objective defined in Eq. (16) and $B$
12:      **end if**
13: **end for**

---

**Assumption 4.3.** Let $\Pi$ be the set of admissible diffusion policies satisfying $\forall p^\pi \in \Pi$, $\forall s \in \mathcal{S}$,

$$\sup_{a^0 \in \mathcal{A}} \frac{\pi(a^0|s)}{\nu(a^0|s)} = \frac{\int p_{0:N}^{\pi,s}(a^{0:N}) \mathrm{d}a^{1:N}}{\int p_{0:N}^{\nu,s}(a^{0:N}) \mathrm{d}a^{1:N}} < \infty,$$

where $p^\nu$ is the behavior diffusion policy.

**Proposition 4.4.** *(Soft Policy Improvement, proof in Appendix C.4) Let $p^{\pi_{new}}$ be the optimizer of the problem defined in Eq. (16). Under Assumption 4.3, $V_n^{\pi_{new},s}(a^n) \geq V_n^{\pi_{old},s}(a^n)$ holds for all $n \in \{0, 1, \ldots, N\}$ and $(s, a) \in \mathcal{S} \times \mathcal{A}$.*

Combining Proposition 4.4 and a similar convergence argument in Soft Actor Critic (Haarnoja et al., 2018), we obtain that the policy is guaranteed to improve w.r.t. the pathwise KL problem in Definition 4.1.

**Proposition 4.5.** *(Policy Iteration, proof in Appendix C.5) Under Assumption 4.3, repeated application of soft policy evaluation in Eq. (13) and Eq. (15) and soft policy improvement in Eq. (16) from any $p^\pi \in \Pi$ converges to a policy $p^{\pi^*}$ such that $V_n^{\pi^*,s}(a) \geq V_n^{\pi,s}(a)$ for all $p^\pi \in \Pi$, $n \in \{0, 1, \ldots, N\}$, and $(s, a) \in \mathcal{S} \times \mathcal{A}$.*

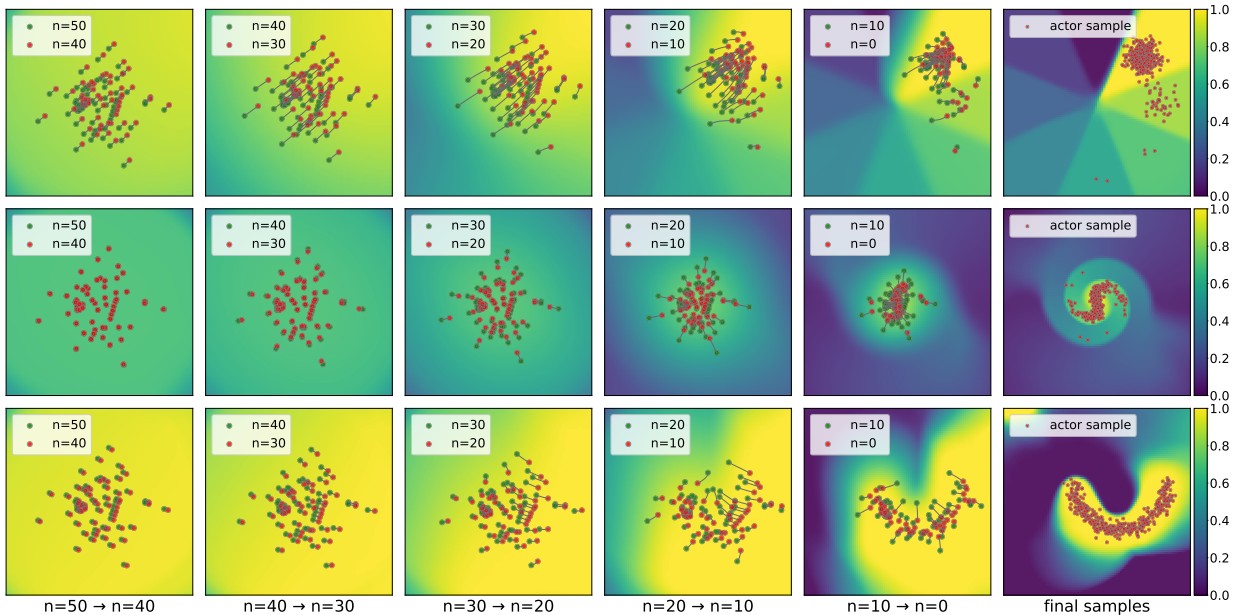

Figure 4. Generation paths of BDPO on the *8gaussian* (top), *2spirals* (middle), and *moons* (down) datasets. The regularization strength is set to $\eta = 0.06$, which is identical to Figure 5. The first five columns depict the diffusion generation process at different time intervals, with green dots indicating the starting points of these intervals, red dots indicating the ending points, and grey lines in between representing intermediate samples. The background color depicts the output from the diffusion value functions $V_n^{\pi,s}$ in the entire 2D space. The rightmost figures depict the final action samples. We use DDIM sampling (Song et al., 2020) for better illustration.

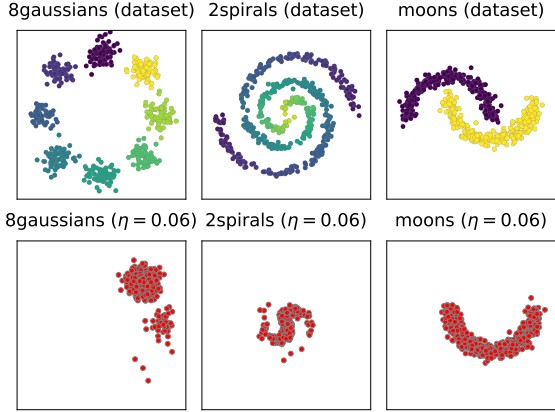

Figure 5. Illustration of the *8gaussians*, *2spirals*, and *moons* datasets. The top row depicts the original data distribution $p_{\text{data}}$, while the second row depicts the target distribution $p_{\text{target}}$ at $\eta = 0.06$ by re-sampling data points according to their energies.

**Remark (Actor-critic across two time scales).** Our treatment of the environment MDP and the (reverse) diffusion MDP resembles DPPO (Ren et al., 2024), which unfolds the diffusion steps into the environment MDP and employs the PPO (Schulman et al., 2017) algorithm to optimize the diffusion policy in the extended MDP. Compared to DPPO, our method adds a penalty in each diffusion step to implement behavior regularization. Besides, we maintain two types of

value functions to account for both inter-timestep and inner-timestep policy evaluation, thereby offering low-variance policy gradients.

### 4.3. Practical Implementation

**Calculation of KL Divergence.** Since each single-step reverse transition $p_{n-1|n}^{\pi,s,a_n}$ is approximately parameterized as an isotropic Gaussian distribution (Eq. (8)), the KL divergence is analytically expressed as

$$D_{\text{KL}}\left[p_{n-1|n}^{\pi,s,a^n}\|p_{n-1|n}^{\nu,s,a^n}\right] = \frac{\|\mu_n^{\pi,s}(a^n) - \mu_n^{\nu,s}(a^n)\|^2}{2\sigma_n^2}. \quad (17)$$

That is, each penalty term is simply the discrepancy between the mean vectors of the reverse transitions, weighted by a factor depending on the noise schedule. The transition of the reverse process becomes Gaussian exactly in the continuous limit, in which case, the KL divergence can be computed through Girsanov's theorem (Oksendal, 2013), and the sum of mean squared errors (MSEs) is replaced by an integral (Franzese et al., 2024). In Appendix D, we demonstrate that the pathwise KL is consistent with the result from Girsanov's theorem in the limit of infinitesimal diffusion step size.

**Selection of States, Diffusion Steps and Actions.** The policy improvement step requires training $V_n^{\pi,s}$ and improving $p^\pi$ on every $(s, a^n, n)$ triplet sampled from the on-policy

*Table 1.* Comparison of `BDPO` and various baseline methods on locomotion-v2 and antmaze-v0 datasets from D4RL. We use 'm' as the abbreviation for medium, 'r' for replay, and 'e' for expert. The performances of the baseline methods are taken from their original paper. For `BDPO`, we report the average and the standard deviation of the performances across 10 evaluation episodes (100 for antmaze datasets) and 5 seeds. We bold values that are within 95% of the top-performing method.

| Dataset | CQL | IQL | DD | SfBC | IDQL-A | QGPO | SRPO | DTQL | Diffusion-QL | EDP | DAC | BDPO (Ours) |
|---|---|---|---|---|---|---|---|---|---|---|---|---|
| halfcheetah-m | 44.0 | 47.4 | 49.1 | 45.9 | 51.0 | 54.1 | 60.4 | 57.9 | 51.1 | 52.1 | 59.1 | **71.2±0.9** |
| hopper-m | 58.5 | 66.3 | 79.3 | 57.1 | 65.4 | **98.0** | 95.5 | **99.6** | 90.5 | 81.9 | **101.2** | **100.6±0.7** |
| walker2d-m | 72.5 | 78.3 | 82.5 | 77.9 | 82.5 | 86.0 | 84.4 | 89.4 | 87.0 | 86.9 | 96.8 | **93.4±0.5** |
| halfcheetah-m-r | 45.5 | 44.2 | 39.3 | 37.1 | 45.9 | 47.6 | 51.4 | 50.9 | 47.8 | 49.4 | 55.0 | **58.9±0.9** |
| hopper-m-r | 95.0 | 94.7 | 100.0 | 86.2 | 92.1 | 96.9 | 101.2 | 100.0 | 101.3 | 101.0 | 103.1 | **101.4±0.5** |
| walker2d-m-r | 77.2 | 73.9 | 75.0 | 65.1 | 85.1 | 84.4 | 84.6 | 88.5 | 95.5 | 94.9 | 96.8 | **95.5±1.6** |
| halfcheetah-m-e | 91.6 | 86.7 | 90.6 | 92.6 | 95.9 | 93.5 | 92.2 | 92.7 | 96.8 | 95.5 | 99.1 | **108.7±0.9** |
| hopper-m-e | **105.4** | 91.5 | **111.8** | 108.6 | 108.6 | 108.0 | 100.1 | 109.3 | 111.1 | 97.4 | **111.7** | **111.3±0.2** |
| walker2d-m-e | 108.8 | **109.6** | 108.8 | **109.8** | 112.7 | 110.7 | 114.0 | 110.0 | 110.1 | 110.2 | 113.6 | **115.6±0.4** |
| **locomotion sum** | 698.5 | 749.7 | 736.2 | 680.3 | 739.2 | 779.2 | 783.8 | 798.3 | 791.2 | 769.3 | 836.4 | **856.7** |
| antmaze-u | 74.0 | 87.5 | - | 92.0 | 94.0 | **96.4** | 90.8 | **94.8** | 93.4 | 94.2 | **99.5** | **98.4±1.5** |
| antmaze-u-div | **84.0** | 62.2 | - | **85.3** | 80.2 | 74.4 | 59.0 | 78.8 | 66.2 | 79.0 | **85.0** | 83.0±12.4 |
| antmaze-m-play | 61.2 | 71.2 | - | 81.3 | 84.5 | 83.6 | 73.0 | 79.6 | 76.6 | 81.8 | 85.8 | **92.0±2.9** |
| antmaze-m-div | 53.7 | 70.0 | - | 82.0 | 84.8 | 83.8 | 65.2 | 82.2 | 78.6 | 82.3 | 84.0 | **93.2±2.8** |
| antmaze-l-play | 15.8 | 39.6 | - | 59.3 | 63.5 | **66.6** | 38.8 | 52.0 | 46.4 | 42.3 | 50.3 | **69.0±8.0** |
| antmaze-l-div | 14.9 | 47.5 | - | 45.5 | 67.9 | 64.8 | 33.8 | 54.0 | 56.6 | 60.6 | 55.3 | **84.0±3.2** |
| **antmaze sum** | 303.6 | 378.0 | - | 445.4 | 474.9 | 469.6 | 360.6 | 441.4 | 417.8 | 440.2 | 459.9 | **519.6** |

distribution of $p^\pi$. However, we employ an off-policy approach, by sampling $(s, a)$ from the dataset, $n$ according to the Uniform distribution $U[1, N]$ and $a^n$ according to $q_{n|0}(\cdot|a)$, which we found works sufficiently well in our experiments. We note that on-policy sampling $(s, a^n, n)$ may further benefit the performance at the cost of sampling and restoring these triplets on the fly, similar to what iDEM (Akhound-Sadegh et al., 2024) did in its experiments.

**Lower-Confidence Bound (LCB) Value Target.** Following existing practices (Zhang et al., 2024a; Fang et al., 2024), we use an ensemble of $K = 10$ value networks $\{\psi_k, \phi_k\}_{i=k}^K$ for both $V_n^{\pi,s}$ and $Q^\pi$ and calculate their target as the LCB of Eq. (13) and Eq. (15):

$$
\begin{aligned}
y_Q &= \mathrm{Avg}_k[\mathcal{B}^\pi Q_{\bar\psi_k}^\pi] - \rho\sqrt{\mathrm{Var}_k[\mathcal{B}^\pi Q_{\bar\psi_k}^\pi]}, \\
y_V &= \mathrm{Avg}_k[\mathcal{T}_n^\pi V_{n,\bar\phi_k}^{\pi,s}] - \rho\sqrt{\mathrm{Var}_k[\mathcal{T}_n^\pi V_{n,\bar\phi_k}^{\pi,s}]},
\end{aligned}
\tag{18}
$$

where $\bar\phi_k$ and $\bar\psi_k$ are exponentially moving average versions of the value networks, and $\mathrm{Avg}$, $\mathrm{Var}$ denote averages and variances respectively. The objectives for value networks are to minimize the mean-squared error between the prediction and the target:

$$
\begin{aligned}
\mathcal{L}(\{\psi_k\}_{k=1}^K) &= \mathbb{E}_{(s,a,s',r)\sim\mathcal{D}}\left[\sum_{k=1}^K (y_Q - Q_{\psi_k}^\pi(s,a))^2\right], \\
\mathcal{L}(\{\phi_k\}_{k=1}^K) &= \mathbb{E}_{(s,a)\sim\mathcal{D},n,a^n}\left[\sum_{k=1}^K (y_V - V_{n,\phi_k}^{\pi,s}(a^n))^2\right],
\end{aligned}
\tag{19}
$$

where $n$ is sampled from $U[1, N]$ and $a^n$ from $q_{n|0}(\cdot|a)$.

The pseudo-code for `BDPO` is provided in Algorithm 1.

## 5. Experiments

We evaluate `BDPO` with synthetic 2D tasks and also the D4RL benchmark (Fu et al., 2020). Due to the space limit, a detailed introduction about these tasks and the hyperparameter configurations is deferred to Section A and Section B. More experimental results are deferred to Appendix E.

### 5.1. Results of Synthetic 2D Datasets

We leverage synthetic 2D datasets used in Lu et al. (2023) to further our understanding of the mechanism of `BDPO`. Each dataset contains data points $x_i$ paired with specific energy values $\mathcal{E}(x_i)$. By re-sampling the data according to the Boltzmann distribution $p(x_i) \propto \exp(\mathcal{E}(x_i)/\eta)$, we can obtain the ground truth target distribution at a specific regularization strength $\eta$ (Haarnoja et al., 2018; Nair et al., 2020):

$$
p_{\text{target}}(x_i) \propto p_{\text{data}}(x_i)\exp(\mathcal{E}(x_i)/\eta).
$$

As an example, Figure 5 illustrates both the original dataset and the re-sampled dataset with $\eta = 0.06$ for the *8gaussians*, *2spirals*, and *moons* datasets.

Our objective is to analyze the properties of the diffusion policy and the diffusion value function trained with `BDPO`. The results are depicted in Figure 4 (full results on all datasets in Figure 13), where each row corresponds to a different

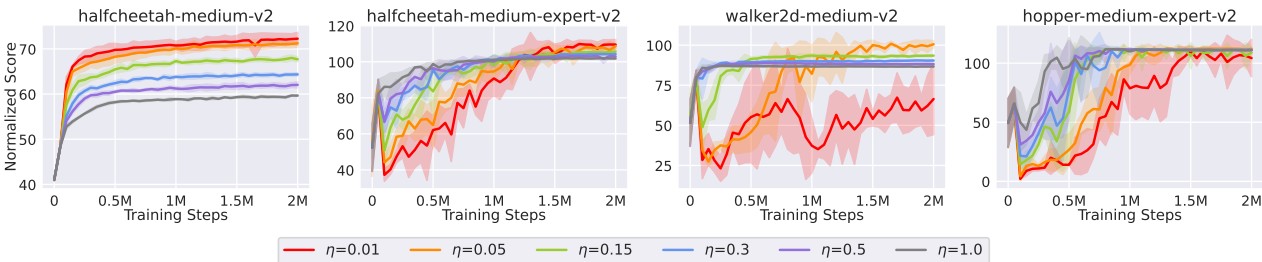

*Figure 6.* Sensitivity analysis of the regularization strength $\eta$. For each configuration, we report the mean and the standard deviation of the performances aggregated from 5 independent seeds and 10 evaluation episodes for each seed.

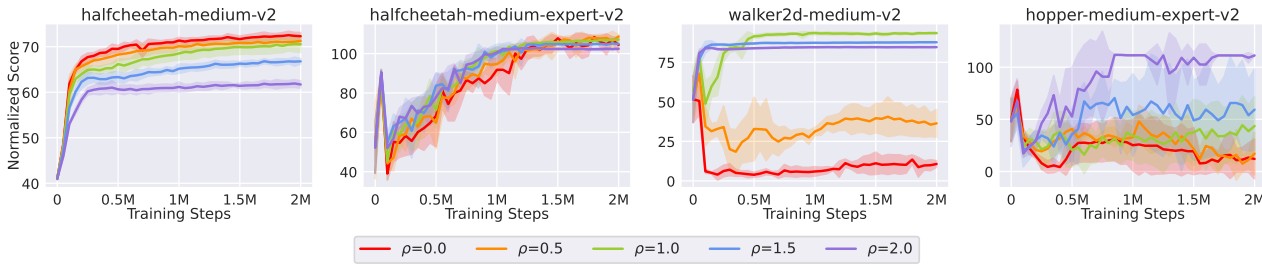

*Figure 7.* Sensitivity analysis of the lower confidence bound coefficient $\rho$. For each configuration, we report the mean and the standard deviation of the performances aggregated from 5 independent seeds and 10 evaluation episodes for each seed.

dataset and visualizes the iterative sampling process. From left to right, the samples evolve from an initial noisy distribution toward a well-defined structure as the reverse diffusion progresses. In the initial steps ($n = 50$ to $n = 40$), the sample movement is subtle, while in the later steps ($n = 30$ to $n = 0$), the samples rapidly converge to the nearest modes of the data. The final column on the right presents the ultimate samples generated by the policy, which closely match the ground-truth target distribution visualized in Figure 5.

Besides the generation result, the background color in Figure 4 depicts the landscapes of the diffusion value functions $V^\pi$ at $n = 50/40/30/20/10/0$. In earlier, noisier steps (e.g., $n = 50$), the output values across the space are similar, resulting in a smoother landscape and weaker guidance during sampling. As the noise level decreases in later steps (e.g., $n = 10$), the value outputs vary more sharply, creating stronger guidance on the diffusion policy. Such progression enables the model to refine samples while exploring sufficiently across the space, finally yielding results that align with the target distribution.

### 5.2. Results on Continuous Control Tasks

To assess the performance of `BDPO` on complex continuous control problems, we utilize datasets from D4RL as our test bench and compare `BDPO` against a wide spectrum of offline RL algorithms. Specifically, we include 1) CQL (Kumar et al., 2020b) and IQL (Kostrikov et al., 2022), which are

representatives using straightforward one-step policies; 2) Decision Diffuser (DD) (Ajay et al., 2023), which uses diffusion models for planning and extracts actions from planned trajectories; 3) SfBC (Chen et al., 2023), IDQL (Hansen-Estruch et al., 2023), and QGPO (Lu et al., 2023), which use diffusion to model the behavior and further refine actions using the value functions; 4) SRPO (Chen et al., 2024a) and DTQL (Chen et al., 2024b), which use diffusion to provide behavior regularization for one-step actors; 5) Diffusion-QL (Wang et al., 2023), which propagates the gradient of Q-value functions over the whole generation process; and 6) EDP (Kang et al., 2023) and DAC (Fang et al., 2024), which accelerate the training of diffusion policies by using action approximation or using proxy objectives. The performance of each algorithm is measured by the normalized score on each task (see Appendix A for details).

The results are listed in Table 1. Our findings indicate that diffusion-based methods, particularly those with diffusion-based actor and regularization (including `BDPO`, DAC, and Diffusion-QL), substantially outperform their non-diffusion counterparts, especially in locomotion tasks. Meanwhile, `BDPO` consistently achieves superior performance across nearly all datasets, underscoring the effectiveness of combining diffusion policies and the behavior-regularized RL framework. Besides the final performance, we plot the training curves of `BDPO` in Section E.4. Overall, `BDPO` achieves fast and stable convergence except for some of the antmaze

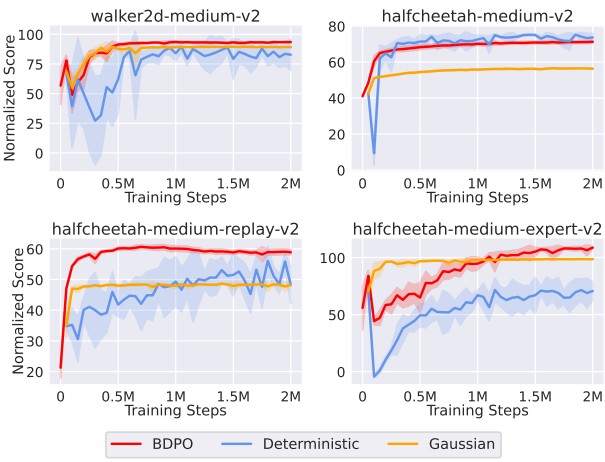

*Figure 8.* Comparison of different policy parameterizations. Results are taken from 10 evaluation episodes and 5 seeds.

*Table 2.* Runtime comparison among `BDPO`, DAC, and Diffusion-QL (denoted as D-QL in the table), measured in minutes.

| Alg | Phase | $N = 5$ | $N = 20$ | $N = 50$ |
|---|---|---|---|---|
| `BDPO` | policy | 17 | 20 | 18 |
| | value | 63 | 135 | 285 |
| | total | 80 | 155 | 303 |
| DAC | policy | 30 | 31 | 30 |
| | value | 30 | 109 | 251 |
| | total | 60 | 140 | 285 |
| D-QL | policy | 57 | 132 | 308 |
| | value | 31 | 80 | 252 |
| | total | 88 | 212 | 560 |

datasets, where variations may occur due to the sparse reward nature of these datasets.

### 5.3. Analysis of `BDPO`

**Regularization Strength $\eta$.** The hyperparameter $\eta$ controls the regularization strength: larger $\eta$ forces the diffusion policy to stay closer to the behavior diffusion, while smaller $\eta$ places more emphasis on the guidance from value networks. Figure 6 demonstrates this effect: in general, smaller $\eta$ does lead to better performances. However, excessively small values can result in performance degradation, as observed in *walker2d-medium-v2*. In contrast to auto-tuning $\eta$ via dual gradient descend (Fang et al., 2024; Zhang et al., 2024a), we find that adjustable $\eta$ causes fluctuations in penalty calculation. Therefore, we employ a fixed $\eta$ throughout training.

**Lower Confidence Bound Coefficient $\rho$.** The hyperparameter $\rho$ determines the level of pessimism on OOD actions since the variances of Q-values on these actions are comparatively higher than in-dataset actions. In Figure 7, we discover that there exists a certain range of $\rho$ where the lower confidence bound value target works best, while excessively larger or smaller values lead to either underestimation or overestimation in value estimation.

**Policy Parameterization.** As discussed in Section 1, improper assumptions about the action distribution may lead to inaccurate penalty calculations and ultimately degrade the overall performance. To validate this claim, we conduct a series of ablation studies that replace the actor parameterization with Gaussian distributions and Dirac distributions, respectively, while keeping other experimental configurations unchanged. For a fair comparison, we fine-tune the regularization strength $\eta$ by searching within a predefined range and report the performance corresponding to the op-

timal value. The results are illustrated in Figure 8, with additional details regarding the implementations provided in Appendix B.3. While simpler architectures generally converge more quickly, their performance at convergence often lags behind that of `BDPO`, indicating a limitation in their capacity for policy parameterization.

**Discussion on Runtime.** Table 2 compares the runtime of `BDPO`, DAC, and Diffusion-QL using the *walker2d-medium-v2* dataset. Our method consists of three key steps: pretraining behavior diffusion, training value functions ($Q^\pi$ and $V_n^{\pi,s}$), and optimizing the actor. The pertaining phase takes around 8 minutes and is therefore omitted in further analysis. The $Q^\pi$ training follows the same approach as Diffusion-QL and DAC, involving diffusion path sampling at the next state, while $V_n^{\pi,s}$ introduces acceptable overhead since it only requires single-step diffusion. For actor training, both DAC and `BDPO` use single-step diffusion, ensuring computational costs remain constant regardless of diffusion steps. In contrast, Diffusion-QL needs to back-propagate through the entire diffusion path, leading to a significantly higher runtime, especially when the number of diffusion steps increases.

### 6. Conclusions

Our core contribution is to extend the behavior-regularized RL framework to diffusion policies by implementing the regularization as the accumulated discrepancies of each single-step transition of the reverse diffusion. Theoretically, we demonstrate the equivalence of the pathwise KL and the commonly adopted KL constraint on action distributions. In practice, we instantiate the framework with an actor-critic style algorithm that leverages value functions on two time scales for efficient optimization. The proposed algorithm, `BDPO`, produces generation results that closely align with the target distribution, thereby enhancing its performance and applicability in complex continuous control problems.

## Acknowledgements

This work is primarily supported by the National Science Foundation of China (62276126, 62250069). Chenjun Xiao gratefully acknowledges funding from the National Natural Science Foundation of China (62406271).

## Impact Statement

This paper presents work whose goal is to advance the field of machine learning. There are many potential societal consequences of our work, none of which we feel must be specifically highlighted here.

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

# A. Introduction about the Benchmarks

**Synthetic 2D Datasets.** We leverage the synthetic 2D datasets from Lu et al. (2023) as a sanity check and a convenient illustration of the mechanism of `BDPO`. This task set encompasses 6 different datasets. For each dataset, the data points $x_i$ are associated with different energies $\mathcal{E}(x_i)$. By sampling from the Boltzmann distribution associated with the temperature $\eta$:

$$p(x_i) \propto \exp(\mathcal{E}(x_i)/\eta), x_i \in \mathcal{D},$$

we can obtain ground-truth sample results at a specific regularization strength $\eta$ (see Figure 9).

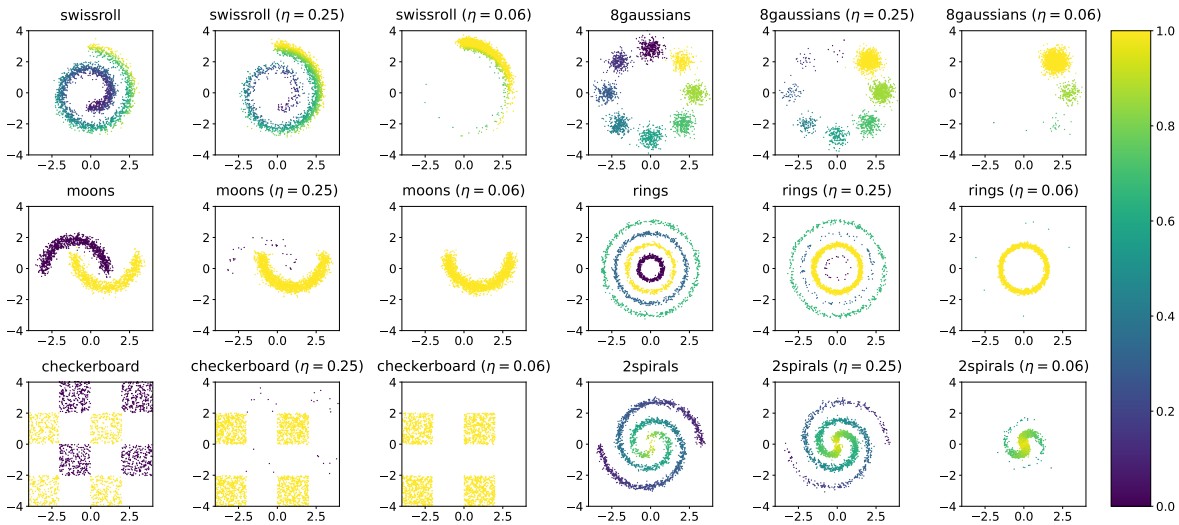

*Figure 9.* Illustration of synthetic 2D datasets and sampling results with various temperature $\eta$.

We also use Gym MuJoCo (Brockman et al., 2016), which provides a series of continuous control tasks to assess the performance of `BDPO` as well as the baseline methods.

**Locomotion Tasks.** We choose 3 tasks from Gym MuJoCo: 1) *halfcheetah*, which is a robot with 9 body segments and 8 joints (including claws) as shown in Figure 10a. The goal is to apply torque to 6 of the joints to make the robot move forward as quickly as possible. The reward is based on forward movement distance. 2) *hopper*, which is a simpler robot with a torso, a thigh, a shin, and a foot as shown in Figure 10b. The goal is to use torque on 3 joints to make the robot hop forward. 3) *walker2d*, a robot with a torso and two legs, each consisting of a thigh, a shin, and a foot as shown in Figure 10c. The goal is to apply torque to 6 joints to make the robot walk forward, rather than hop.

For the offline dataset, we choose the *-v2* datasets with three levels of qualities provided by D4RL (Fu et al., 2020): 1) *medium*, which is collected by a policy that achieves approximately 30% of expert-level performance; 2) *medium-replay*, which includes all data from the replay buffer of the training process of the medium-level policy; and 3) *medium-expert*: a mixture of medium-level data and expert-level data in a 1:1 ratio.

**Navigation Tasks.** We use *antmaze* as the representative of navigation tasks. In this task set, the agent needs to control an eight degree-of-freedom *ant* robot to walk through a maze, as shown in Figure 10d. We use three different maps: *umaze*, *medium*, and *large*; and employed three different types of datasets: 1) the ant reaches a fixed target from a fixed starting position; 2) In the *diverse* dataset, the ant needs to go from a random position to a random target; 3) In the *play* dataset, the ant goes from manually selected positions to manually selected targets (not necessarily the same as during evaluation). We use the *-v0* versions of the datasets, also provided by D4RL (Fu et al., 2020).

**Metrics.** We measure the performance using the normalized score, which can be calculated by

$$\text{Normalized\_Score}(\pi) = \frac{\text{Score}(\pi) - \text{Score}_{\text{random}}}{\text{Score}_{\text{expert}} - \text{Score}_{\text{random}}},$$

where $\text{Score}(\pi)$ is the average cumulated rewards achieve by the policy $\pi$, $\text{Score}_{\text{random}}$ and $\text{Score}_{\text{expert}}$ are reference scores for random policies and expert policies, provided by D4RL.

# B. Hyper-parameter Configurations

### B.1. D4RL Datasets

We largely follow the configurations from DAC (Fang et al., 2024). Parameters that are common across all datasets are listed in Table 3. The hyperparameters of value networks in the table are the same for both the Q-value networks $Q_\psi$ and diffusion value networks $V_\phi$.

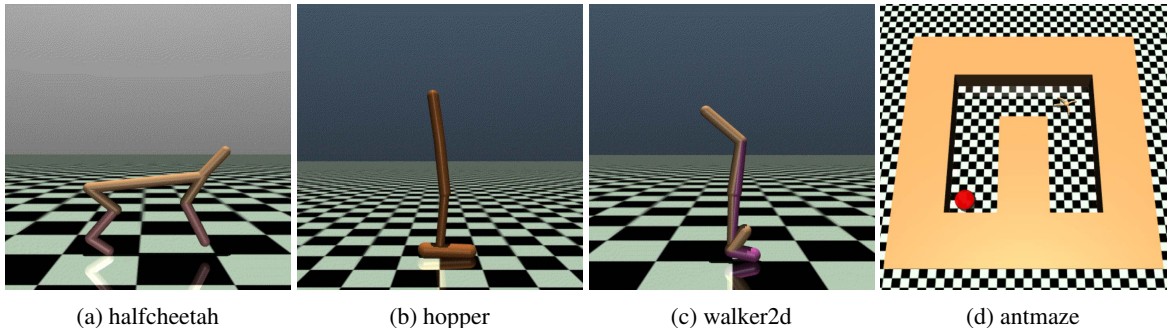

| (a) halfcheetah | (b) hopper | (c) walker2d | (d) antmaze |

*Figure 10.* Illustration of the Locomotion and Navigation Tasks.

*Table 3.* Common hyperparameters across all datasets.

| | |
|---|---|
| Diffusion Backbone | MLP(256, 256, 256, 256) for locomotion tasks
MLP(512, 512, 512, 512) for antmaze tasks |
| Diffusion Noise Schedule | Variance Preserving (Song et al., 2021) |
| Diffusion Solver | DDPM |
| Diffusion Steps | 5 |
| Behavior Learning Rate | 0.0003 for locomotion tasks and 0.0001 for antmaze tasks |
| Actor Learning Rate | 1e-5 |
| Actor Learning Rate Schedule | Cosine Annealing |
| Actor Gradient Clip | 1.0 |
| Actor EMA Rate | 5e-3 |
| Actor Sampling | For *antmaze-umaze-diverse-v0*, generate 1 sample and execute
For others, generate 10 samples and select the action with the highest Q |
| Actor Update Frequency | 5 |
| Value Learning Rate | 0.0003 |
| Value Ensemble | 10 (20 for the *hopper-medium-v2* task exclusively) |
| Value Backbone | MLP(256, 256, 256) |
| Value EMA Rate | 0.005 |
| Discount | 0.99 for locomotion tasks and 0.995 for antmaze tasks |
| Batch Size | 256 |
| Optimizer | ADAM (Kingma & Ba, 2015) |
| Pretrain Steps | 2e6 |
| Train Steps | 2e6 |
| Value Warmup Steps | 5e4 |
| Sample Clip Range | [-1, 1] |

We list the hyper-parameters that vary across different datasets in Table 4. We specifically tune the regularization strength $\eta$ and the LCB coefficient for locomotion tasks, plus the value backup mode for antmaze tasks.

For evaluation, at each timestep $s_t$, we use the diffusion policy to diffuse $N = 10$ actions, and select the action with the highest Q-value as the final action $a_t$ to execute in the environment. We report the score normalized by the performances of random policies and expert policies, provided by D4RL (Fu et al., 2020).

*Table 4.* Hyper-parameters that vary in different tasks.

| Dataset | $\rho$ | $\eta$ | Max-Q Backup (Wang et al., 2023) |
|---|---|---|---|
| halfcheetah-medium-v2
halfcheetah-medium-replay-v2
halfcheetah-medium-expert-v2 | 0.5 | 0.05 | False |
| hopper-medium-v2
hopper-medium-replay-v2
hopper-medium-expert-v2 | 2.0 | 0.2 | False |
| walker2d-medium-v2
walker2d-medium-replay-v2
walker2d-medium-expert-v2 | 1.0 | 0.15 | False |
| antmaze-umaze-v2
antmaze-umaze-diverse-v0 | 0.8 | 0.5 | True |
| antmaze-medium-play-v0
antmaze-medium-diverse-v0 | 0.8 | 0.2 | True |
| antmaze-large-play-v0
antmaze-large-diverse-v0 | 0.8 | 1.0 | True |

## B.2. Synthetic 2D Datasets

In the experiments with Synthetic 2D Datasets, we basically followed the parameters in D4RL in Table 3, except that $\rho$ is set to 0, $\eta$ is set to 0.06, batch size is set to 512, and the number of diffusion steps is set to 50. We also turn off sample clipping for BDPO since the actions in these tasks are within $[-4.5, 4.5]$.

## B.3. Details about the Ablation on Policy Parameterization

In this section, we provide more details about how we implement the variants that use deterministic policies or Gaussian policies. The only difference between these variants and `BDPO` lines in the actor parameterization, which means that we keep the implementation of the value networks (e.g., ensemble tricks and LCB value target) and the hyperparameters (e.g., the LCB coefficient $\rho$) identical to `BDPO`. We inherit the general behavior-regularized RL framework defined in Eq. (1), where we update the soft Q-value functions using the target

$$\mathcal{B}^\pi Q^\pi(s, a) = R(s, a) + \gamma \mathbb{E}\left[Q^\pi(s', a')\right] - \eta D(\pi, \nu), \tag{20}$$

and improve the policy $\pi$ via

$$\max_\pi \ \gamma \mathbb{E}_{a \sim \pi}\left[Q^\pi(s, a)\right] - \eta D(\pi, \nu), \tag{21}$$

where $D$ denotes certain types of regularization specific to the actor parameterization.

For deterministic policies, we follow Tarasov et al. (2023) and use the mean squared error between actions from the policy and the behavior policy:

$$D(\pi, \nu) = \frac{1}{2}\|\pi(s) - \nu(s)\|^2. \tag{22}$$

For Gaussian policies, we parameterize it as Diagonal Gaussian distribution $\pi(\cdot|s) = \mathcal{N}(\mu_\pi(s), \sigma_\pi^2(s))$, and the regularization is implemented as the KL divergence:

$$D(\pi(s), \nu(s)) = D_{\text{KL}}\left[\pi(s)\|\nu(s)\right] = \log\frac{\sigma_\nu(s)}{\sigma_\pi(s)} + \frac{\sigma_\pi^2(s) + (\mu_\pi(s) - \mu_\nu(s))^2}{2\sigma_\nu^2(s)} - \frac{1}{2}. \tag{23}$$

We find that using the analytical KL computation for Gaussian policies works better than using sample-based estimations.

To ensure fair comparisons, we search for the optimal $\eta$ for both deterministic policies and Gaussian policies within a certain range. Specifically, we choose the $\eta$ from $\{0.1, 0.25, 0.5, 0.75, 1, 2, 3, 5, 10\}$. We observed that a higher eta would lead to a

decline in performance, but an excessively small eta would cause the performance to oscillate drastically or even collapse in some cases. Therefore, we select values that ensure stable performance and deliver the highest level of performance as shown in Table 5.

*Table 5.* Values of $\eta$ used in the ablation study with the policy parameterization.

| Dataset | Deterministic Policy | Gaussian Policy |
|---|---|---|
| halfcheetah tasks | 0.25 | 0.75 |
| hopper tasks | 1.0 | 0.1 |
| walker2d tasks | 1.5 | 0.25 |

# C. Proof for the Main Results

**Theorem C.1.** *Suppose that the behavior policy is a diffusion model that approximates the forward process $q$ by minimizing the objective*

$$\mathbb{E}_{a^0 \sim \nu(s), a^n \sim q_{n|0}(\cdot|a^0)} \left[ D_{\mathrm{KL}} \left[ q_{n-1|0,n}(\cdot|a^0, a^n) \| p^{\nu,s}_{n-1|n}(\cdot|a^n) \right] \right] \quad \forall n \in \{1, 2, \ldots, N\}, s \in \mathcal{S}. \tag{24}$$

*Then, for any given Q-value function $Q(s, a)$, the optimal solution $p^{*,s}$ of Eq. (14) is the reverse process corresponding to the forward process $q^{\pi_Q,s}_{0:N} = \pi_Q(a^0|s) \prod_{n=1}^{N} q_{n|n-1}(a^n|a^{n-1})$, where $\pi_Q(a|s) \propto \nu(a|s) \exp(Q(s,a)/\eta)$.*

*Proof.* Let's first consider the behavior diffusion policy $p^\nu$, which minimizes the following problem for every $n$ and state $s$:

$$\min_{p^{\nu,s}_{n-1|n}} \quad \mathbb{E}_{a^0 \sim \nu(\cdot|s)} \left[ q_{n|0}(a^n|a^0) D_{\mathrm{KL}} \left[ q_{n-1|0,n}(\cdot|a^0, a^n) \| p^{\nu,s}_{n-1|n}(\cdot|a^n) \right] \right]$$

$$\text{s.t.} \quad \int p^{\nu,s}_{n-1|n}(a^{n-1}|a^n) \mathrm{d}a^{n-1} = 1.$$

The corresponding Lagrange function is

$$L(p^{\nu,s}_{n-1|n}, \lambda) = \mathbb{E}_{a^0 \sim \nu(\cdot|s)} \left[ q_{n|0}(a^n|a^0) D_{\mathrm{KL}} \left[ q_{n-1|0,n}(\cdot|a^0, a^n) \| p^{\nu,s}_{n-1|n}(\cdot|a^n) \right] \right] + \lambda(1 - \int p^{\nu,s}_{n-1|n}(a^{n-1}|a^n) \mathrm{d}a^{n-1}), \tag{25}$$

where $\lambda$ is the multiplier. Setting $\frac{\partial L}{\partial p^{\nu,s}_{n|n-1}} = 0$, we obtain

$$p^{\nu,s}_{n-1|n}(a^{n-1}|a^n) = -\frac{1}{\lambda} \mathbb{E}_{a^0 \sim \nu^0(\cdot|s)} \left[ q_{n|0}(a^n|a^0) q_{n-1|0,n}(a^{n-1}|a^0, a^n) \right]. \tag{26}$$

Since $\int p^{\nu,s}_{n-1|n}(a^{n-1}|a^n) \mathrm{d}a^{n-1} = 1$, it follows that

$$\begin{aligned}
\lambda &= -\int \mathbb{E}_{a^0 \sim \nu(\cdot|s)} \left[ q_{n|0}(a^n|a^0) q_{n-1|0,n}(a^{n-1}|a^0, a^n) \right] \mathrm{d}a^{n-1} \\
&= -\mathbb{E}_{a^0 \sim \nu(\cdot|s)} \left[ q_{n|0}(a^n|a^0) \right].
\end{aligned} \tag{27}$$

Substituting Eq. (27) into Eq. (26),

$$\begin{aligned}
p^{\nu,s}_{n-1|n}(a^{n-1}|a^n) &= \frac{\mathbb{E}_{a^0 \sim \nu(\cdot|s)} \left[ q_{n|0}(a^n|a^0) q_{n-1|0,n}(a^{n-1}|a^0, a^n) \right]}{\mathbb{E}_{a^0 \sim \nu^0(\cdot|s)} \left[ q_{n|0}(a^n|a^0) \right]} \\
&= \frac{\mathbb{E}_{a^0 \sim \nu(\cdot|s)} \left[ q_{n|0}(a^n|a^0) q_{n-1|0,n}(a^{n-1}|a^0, a^n) \right]}{q_n(a^n)} \\
&= \int \frac{v(a^0) q_{n|0}(a^n|a^0) q_{n-1|0,n}(a^{n-1}|a^0, a^n)}{q_n(a^n)} \mathrm{d}a^0 \\
&= q^{\nu,s}_{n-1|n}(a^{n-1}|a^n),
\end{aligned} \tag{28}$$

That is, through optimizing Eq. (24), the derived behavior diffusion policy is consistent with $q_{n-1|n}^{\nu,s}$ from the forward process.

Next, we prove that the optimizer of Eq. (14) also satisfies $p_{n-1|n}^{*,s} = q_{n-1|n}^{\pi_Q,s}$.

To do this, we first recognize that the reverse (generative) process of the diffusion policy constitutes a Markov Decision Process (MDP), where the *state* at time step $n$ is the tuple $(s, a^n)$ and the agent's policy follows $a^{n-1} \sim p_{n-1|n}^{\pi,s}(\cdot|a^n)$. The *transition* in this MDP is implicit and deterministic: upon selecting the action $a^{n-1}$, the state immediately transitions from $(s, a^n)$ to $(s, a^{n-1})$. At the last time step $n = 0$, the agent will receive an ending reward of $Q(s, a^0)$. In this sense, the problem defined in Eq. (14) can be recognized as a behavior-regularized RL problem with Gaussian parameterized policy $p_{n-1|n}^{\pi,s}$ and behavior policy $p_{n-1|n}^{\nu,s}$. Following Wu et al. (2019), the optimal diffusion value functions satisfy the recursion:

$$
\begin{aligned}
V_0^{*,s}(a^0) &= Q(s, a^0), \\
V_n^{*,s}(a^n) &= \eta \log \mathbb{E}_{a^{n-1} \sim p_{n-1|n}^{\nu,s}} \left[ \exp(V_{n-1}^{*,s}(a^{n-1})/\eta) \right],
\end{aligned}
\tag{29}
$$

and the optimal diffusion policy is given by:

$$
p_{n-1|n}^{*,s}(a^{n-1}|a^n) = \frac{p_{n-1|n}^{\nu,s}(a^{n-1}|a^n) \exp(V_{n-1}^{*,s}(a^{n-1})/\eta)}{\mathbb{E}_{a^{n-1} \sim p_{n-1|n}^{\nu,s}} \left[ \exp(V_{n-1}^{*,s}(a^{n-1})/\eta) \right]}.
\tag{30}
$$

Expanding the recursion of diffusion value functions, we have

$$
\begin{aligned}
V_n^{*,s}(a^n) &= \eta \log \mathbb{E}_{a^{n-1} \sim p_{n-1|n}^{\nu,s}} \left[ \exp(V_{n-1}^{*,s}(a^{n-1})/\eta) \right] \\
&= \eta \log \mathbb{E}_{a^{n-1} \sim p_{n-1|n}^{\nu,s}, a^{n-2} \sim p_{n-2|n-1}^{\nu,s}} \left[ \exp(V_{n-2}^{*,s}(a^{n-2})/\eta) \right] \\
&= \ldots \\
&= \eta \log \mathbb{E}_{a^0 \sim p_{0|n}^{\nu,s}} \left[ \exp(V_0^{*,s}(a^0)/\eta) \right] \\
&= \eta \log \mathbb{E}_{a^0 \sim p_{0|n}^{\nu,s}} \left[ \exp(Q(s, a^0)/\eta) \right].
\end{aligned}
\tag{31}
$$

Similarly, the optimal policy can be expressed as:

$$
p_{n-1|n}^{*,s}(a^{n-1}|a^n) = \frac{p_{n-1|n}^{\nu,s}(a^{n-1}|a^n) \mathbb{E}_{a^0 \sim p_{0|n-1}^{\nu,s}} \left[ \exp(Q(s, a^0)/\eta) \right]}{\mathbb{E}_{a^{n-1} \sim p_{n-1|n}^{\nu,s}} \left[ \mathbb{E}_{a^0 \sim p_{0|n-1}^{\nu,s}} \left[ \exp(Q(s, a^0)/\eta) \right] \right]}.
\tag{32}
$$

Let's consider the following expression:

$$
\begin{aligned}
\mathbb{E}_{a^0 \sim p_{0|n}^{\nu,s}} \left[ \exp(Q(s, a^0)/\eta) \right] &= \int p_{0|n}^{\nu,s}(a^0|a^n) \exp(Q(s, a^0)/\eta) \mathrm{d}a^0 \\
&= \int q_{0|n}^{\nu,s}(a^0|a^n) \exp(Q(s, a^0)/\eta) \mathrm{d}a^0 \\
&= \int \frac{q_{n|0}^{\nu,s}(a^n|a^0)\nu(a^0|s)}{q_n^{\nu,s}(a^n)} \exp(Q(s, a^0)/\eta) \mathrm{d}a^0 \\
&= \frac{1}{Z(s)} \int \frac{q_{n|0}^{\nu,s}(a^n|a^0)\pi_Q(a^0|s)}{q_n^{\nu,s}(a^n)} \mathrm{d}a^0 \\
&= \frac{1}{Z(s)q_n^{\nu,s}(a^n)} \mathbb{E}_{a^0 \sim \pi_Q} \left[ q_{n|0}^{\nu,s}(a^n|a^0) \right],
\end{aligned}
\tag{33}
$$

where $Z(s)$ is the normalizing factor, and the second equation follows from $p_{n-1|n}^{\nu,s} = q_{n-1|n}^{\nu,s}$. Substituting

$\mathbb{E}_{a^0 \sim p^{\nu,s}_{0|n}}\left[\exp(Q(s,a^0)/\eta)\right]$ and $\mathbb{E}_{a^0 \sim p^{\nu,s}_{0|n-1}}\left[\exp(Q(s,a^0)/\eta)\right]$ with Eq. (33) into Eq. (32),

$$
\begin{aligned}
p^{*,s}_{n-1|n}(a^{n-1}|a^n) &= \frac{p^{\nu,s}_{n-1|n}(a^{n-1}|a^n)q^{\nu,s}_n(a^n)\mathbb{E}_{a^0 \sim \pi_Q}\left[q^{\nu,s}_{n-1|0}(a^{n-1}|a^0)\right]}{q^{\nu,s}_{n-1}(a^{n-1})\mathbb{E}_{a^0 \sim \pi_Q}\left[q^{\nu,s}_{n|0}(a^n|a^0)\right]} \\
&= \frac{q^{\nu,s}_{n-1|n}(a^{n-1}|a^n)q^{\nu,s}_n(a^n)\mathbb{E}_{a^0 \sim \pi_Q}\left[q^{\nu,s}_{n-1|0}(a^{n-1}|a^0)\right]}{q^{\nu,s}_{n-1}(a^{n-1})\mathbb{E}_{a^0 \sim \pi_Q}\left[q^{\nu,s}_{n|0}(a^n|a^0)\right]} \\
&= \frac{q^{\nu,s}_{n|n-1}(a^n|a^{n-1})\mathbb{E}_{a^0 \sim \pi_Q}\left[q^{\nu,s}_{n-1|0}(a^{n-1}|a^0)\right]}{\mathbb{E}_{a^0 \sim \pi_Q}\left[q^{\nu,s}_{n|0}(a^n|a^0)\right]} \\
&= \frac{q^{\pi_Q,s}_{n|n-1}(a^n|a^{n-1})\mathbb{E}_{a^0 \sim \pi_Q}\left[q^{\pi_Q,s}_{n-1|0}(a^{n-1}|a^0)\right]}{\mathbb{E}_{a^0 \sim \pi_Q}\left[q^{\pi_Q,s}_{n|0}(a^n|a^0)\right]} \\
&= q^{\pi_Q,s}_{n-1|n}(a^{n-1}|a^n),
\end{aligned}
\tag{34}
$$

where the second line follows from $p^{\nu,s}_{n-1|n} = q^{\nu,s}_{n-1|n}$ and the fourth line follows from the fact that $q^{\nu,s}_{i|j} = q^{\pi_Q,s}_{i|j}$ for any $i > j$.

In conclusion, we obtain $p^{*,s}_{n-1|n}(a^{n-1}|a^n) = q^{\pi_Q,s}_{n-1|n}(a^{n-1}|a^n)$, meaning that the reverse process $p^{*,s}$ exactly corresponds to the forward process starting from $\pi_Q(a|s) \propto \nu(a|s)\exp(Q(s,a)/\eta)$.

$\square$

**Theorem C.2** (Theorem 4.2 in the main text). *Let $p^\nu$ be the behavior diffusion process. The optimal diffusion policy $p^*$ of the pathwise KL-regularized RL problem in Eq. (12) is also the optimal policy $\pi^*$ of the KL regularized objective in Eq. (1), in the sense that $\pi^*(a|s) = \int p^{*,s}_{0:N}(a^{0:N})\delta(a-a^0)\mathrm{d}a^{0:N} \ \forall s \in \mathcal{S}$, where $\delta$ is the Dirac delta function.*

*Proof.* Denote the optimal Q-value function of the pathwise KL-regularized RL as $Q^*$. According to Theorem C.1, for any state $s \in \mathcal{S}$, $p^{*,s}$ is the reverse process of the forward diffusion process $q^{\pi_{Q^*},s}$. Since the $p^{\nu,s}$ is also the reverse process of $q^{\nu,s}$, we have

$$
D_{\mathrm{KL}}\left[p^{*,s}_{0:N}\|p^{\nu,s}_{0:N}\right] = D_{\mathrm{KL}}\left[q^{\pi_{Q^*},s}_{0:N}\|q^{\nu,s}_{0:N}\right].
\tag{35}
$$

Notice that $q^{\pi_{Q^*},s}_{0:N}$ and $q^{\nu,s}_{0:N}$ share the same transition kernel and differ only in the initial distribution. Hence,

$$
\begin{aligned}
D_{\mathrm{KL}}\left[p^{*,s}_{0:N}\|p^{\nu,s}_{0:N}\right] &= D_{\mathrm{KL}}\left[q^{\pi_{Q^*},s}_{0:N}\|q^{\nu,s}_{0:N}\right] \\
&= D_{\mathrm{KL}}\left[q^{\pi_{Q^*},s}_0\|q^{\nu,s}_0\right] + D_{\mathrm{KL}}\left[q^{\pi_{Q^*},s}_{1:N|0}\|q^{\nu,s}_{1:N|0}\right] \\
&= D_{\mathrm{KL}}\left[q^{\pi_{Q^*},s}_0\|q^{\nu,s}_0\right] \\
&= D_{\mathrm{KL}}\left[\pi_{Q^*}(\cdot|s)\|\nu(\cdot|s)\right].
\end{aligned}
\tag{36}
$$

Therefore, the optimal Q-value function $Q^*(s,a)$ satisfies

$$
\begin{aligned}
Q^*(s,a) &= R(s,a) + \gamma\mathbb{E}_{a'^{0:N} \sim p^{*,s'}_{0:N}}\left[Q^*(s',a'0) - \eta D_{\mathrm{KL}}\left[p^{*,s'}_{0:N}\|p^{\nu,s'}_{0:N}\right]\right] \\
&= R(s,a) + \gamma\mathbb{E}_{a' \sim \pi_{Q^*}}\left[Q^*(s',a') - \eta D_{\mathrm{KL}}\left[\pi_{Q^*}(\cdot|s')\|\nu(\cdot|s')\right]\right] \\
&= R(s,a) + \gamma\eta\log\mathbb{E}_{a' \sim \nu(\cdot|s')}\left[\exp(Q^*(s',a')/\eta)\right],
\end{aligned}
\tag{37}
$$

which is exactly the optimal Bellman iteration of the problem defined in Eq. (1). This means that $Q^*$ is also the optimal Q-value function of the problem defined in Eq. (1), and $\pi_{Q^*}$ is correspondingly the optimal solution $\pi^*$. Using Theorem C.1, we know that $p^{*,s}$ and $\pi^*$ are equivalent:

$$
\pi^*(a|s) = \int p^{*,s}_{0:N}(a^{0:N})\delta(a-a^0)\mathrm{d}a^{0:N} \quad \forall s \in \mathcal{S}.
\tag{38}
$$

The proof is completed. $\square$

**Lemma C.3** (Soft Policy Evaluation (adapted from Haarnoja et al. (2018))). *Consider the soft Bellman backup operator* $\mathcal{B}^\pi$ *in Eq. (13) and a function* $Q^0 : \mathcal{S} \times \mathcal{A} \to \mathbb{R}$. *Define* $Q^{k+1} = \mathcal{B}^\pi Q^k$. *Then the sequence* $Q^k$ *will converge to the soft Q-value of* $\pi$ *as* $k \to \infty$ *under Assumption 4.3.*

*Proof.* Define the KL-augmented reward as

$$\tilde{R}^\pi(s, a) := R(s, a) - \eta \mathbb{E}_{a'^{0:N} \sim p_{0:N}^{\pi, s'}} \left[ \sum_{n=1}^{N} D_{\mathrm{KL}} \left[ p_{n-1|n}^{\pi, s'} \| p_{n-1|n}^{\nu, s'} \right] \right],$$

and rewrite the policy evaluation update rule as

$$Q^{k+1}(s, a) \leftarrow \tilde{R}^\pi(s, a) + \gamma \mathbb{E}_{a'^{0:N} \sim p_{0:N}^{\pi, s'}} \left[ Q^k(s', a'^0) \right].$$

When $\gamma \in (0, 1)$, this update satisfies the $\gamma$-contraction property, which is demonstrated in Haarnoja et al. (2018) to converge to a unique solution. Since $Q^\pi$ satisfies $Q^\pi = \mathcal{B}^\pi Q^\pi$, we obtain that the sequence $Q^k$ converges to $Q^\pi$. Note that we employ a bounded probability ratio to ensure the boundedness of the $Q$-value function rather than requiring the action space to be a finite set. $\qquad\square$

**Proposition C.4** (Soft Policy Improvement). *Let* $p^{\pi_{new}}$ *be the optimizer of the problem defined in Eq. (16). Under Assumption 4.3,* $V_n^{\pi_{new}, s}(a^n) \geq V_n^{\pi_{old}, s}(a^n)$ *holds for all* $n \in \{0, 1, \ldots, N\}$ *and* $(s, a) \in \mathcal{S} \times \mathcal{A}$.

*Proof.* Since $p^{\pi_{new}, s, a^n}$ is the optimizer of the problem 16, it satisfies

$$-\eta \ell_n^{\pi_{old}, s}(a^n) + \mathbb{E}_{p_{n-1|n}^{\pi_{old}, s, a^n}} \left[ V_{n-1}^{\pi_{old}, s}(a^{n-1}) \right] = V_n^{\pi_{old}, s}(a^n) \leq -\eta \ell_n^{\pi_{new}, s}(a^n) + \mathbb{E}_{p_{n-1|n}^{\pi_{new}, s, a^n}} \left[ V_{n-1}^{\pi_{old}, s}(a^{n-1}) \right].$$

For $n = 0$, recursively applying the above inequality leads to

$$V_0^{\pi_{old}, s}(a^0)$$
$$= Q^{\pi_{old}}(s, a^0)$$
$$= R(s, a^0) + \gamma \mathbb{E}_{a'^{0:N} \sim p_{0:N}^{\pi_{old}, s'}} \left[ Q^{\pi_{old}}(s, a'^0) - \eta \sum_{i=1}^{N} \ell_i^{\pi_{old}, s'}(a'^i) \right]$$
$$= R(s, a^0) + \gamma \mathbb{E}_{a'^{0:N} \sim p_{0:N}^{\pi_{old}, s'}} \left[ V_0^{\pi_{old}, s'}(a'^0) - \eta \sum_{i=1}^{N} \ell_i^{\pi_{old}, s'}(a'^i) \right]$$
$$= R(s, a^0) + \gamma \mathbb{E}_{a'^N \sim p_N^{\pi_{old}, s'}} \left[ V_N^{\pi_{old}, s'}(a'^N) \right]$$
$$= R(s, a^0) + \gamma \mathbb{E}_{a'^N \sim p_N^{\pi_{new}, s'}, a'^{N-1} \sim p_{N-1|N}^{\pi_{old}, s'}} \left[ V_{N-1}^{\pi_{old}, s'}(a'^{N-1}) - \eta \ell_N^{\pi_{old}, s'}(a'^N) \right]$$
$$\leq R(s, a^0) + \gamma \mathbb{E}_{a'^N \sim p_N^{\pi_{new}, s'}, a'^{N-1} \sim p_{N-1|N}^{\pi_{new}, s'}} \left[ V_{N-1}^{\pi_{old}, s'}(a'^{N-1}) - \eta \ell_N^{\pi_{new}, s'}(a'^N) \right]$$
$$= R(s, a^0) + \gamma \mathbb{E}_{a'^{N-1:N} \sim p_{N-1:N}^{\pi_{new}, s'}, a'^{N-2} \sim p_{N-2|N-1}^{\pi_{old}, s'}} \left[ V_{N-2}^{\pi_{old}, s'}(a'^{N-2}) - \eta \ell_{N-1}^{\pi_{old}, s'}(a'^{N-1}) - \eta \ell_N^{\pi_{new}, s'}(a'^N) \right]$$
$$\leq R(s, a^0) + \gamma \mathbb{E}_{a'^{N-1:N} \sim p_{N-1:N}^{\pi_{new}, s'}, a'^{N-2} \sim p_{N-2|N-1}^{\pi_{new}, s'}} \left[ V_{N-2}^{\pi_{old}, s'}(a'^{N-2}) - \eta \ell_{N-1}^{\pi_{new}, s'}(a'^{N-1}) - \eta \ell_N^{\pi_{new}, s'}(a'^N) \right]$$
$$\cdots$$
$$\leq R(s, a^0) + \gamma \mathbb{E}_{a'^{0:N} \sim p_{0:N}^{\pi_{new}, s'}} \left[ V_0^{\pi_{old}, s'}(a'^0) - \eta \sum_{i=0}^{N} \ell_i^{\pi_{new}, s'}(a'^i) \right]$$
$$\cdots$$
$$\leq V_0^{\pi_{new}, s}(a^0),$$

where the fifth line follows from the definition of the diffusion value functions, the seventh line follows from the above inequality, and the last line follows from recursively expanding the values of successor states. For $n \in \{1, 2, \ldots, N\}$, the

proof is similar:

$$
\begin{aligned}
&V_n^{\pi_{\text{old}},s}(a^n) \\
&= -\eta \ell_n^{\pi_{\text{old}},s}(a^n) + \mathbb{E}_{p_{n-1|n}^{\pi_{\text{old}},s,a^n}} \left[ V_{n-1}^{\pi_{\text{old}},s}(a^{n-1}) \right] \\
&\leq -\eta \ell_n^{\pi_{\text{new}},s}(a^n) + \mathbb{E}_{p_{n-1|n}^{\pi_{\text{new}},s,a^n}} \left[ V_{n-1}^{\pi_{\text{old}},s}(a^{n-1}) \right] \\
&\leq -\eta \ell_n^{\pi_{\text{new}},s}(a^n) + \mathbb{E}_{p_{n-1|n}^{\pi_{\text{new}},s,a^n}} \left[ -\eta \ell_{n-1}^{\pi_{\text{new}},s}(a^{n-1}) + \mathbb{E}_{p_{n-2|n-1}^{\pi_{\text{new}},s,a^{n-1}}} \left[ V_{n-2}^{\pi_{\text{old}},s}(a^{n-2}) \right] \right] \\
&\cdots \\
&\leq \mathbb{E}_{p_{0:n-1|n}^{\pi_{\text{new}},s,a^n}} \left[ -\eta \sum_{i=1}^{n} \ell_i^{\pi_{\text{new}},s}(a^i) + V_0^{\pi_{\text{old}},s}(a^0) \right] \\
&\leq \mathbb{E}_{p_{0:n-1|n}^{\pi_{\text{new}},s,a^n}} \left[ -\eta \sum_{i=1}^{n} \ell_i^{\pi_{\text{new}},s}(a^i) + V_0^{\pi_{\text{new}},s}(a^0) \right] \\
&= V_n^{\pi_{\text{new}},s}(a^n)
\end{aligned}
$$

where the last inequality is obtained previously. This completes the proof. Note that the $n = 0$ case already guarantees the policy improvement in the environment MDP. □

**Proposition C.5** (Soft Policy Iteration (adapted from Haarnoja et al. (2018))). *Under Assumption 4.3, repeated application of soft policy evaluation in Eq. (13) and Eq. (15) and soft policy improvement in Eq. (16) from any $p^\pi \in \Pi$ converges to a policy $p^{\pi^*}$ such that $V_n^{\pi^*,s}(a) \geq V_n^{\pi,s}(a)$ for all $p^\pi \in \Pi$, $n \in \{0, 1, \ldots, N\}$, and $(s, a) \in \mathcal{S} \times \mathcal{A}$.*

*Proof.* Let $p^{\pi_i}$ be the diffusion policy at iteration $i$. By Proposition C.4, the sequence $V_n^{\pi_i,s}$ is monotonically increasing. Given that $V_n^{\pi,s}$ is finite due to the boundedness of rewards and the pathwise KL, the sequence converges to some $p^{\pi^*}$. Since the policy improvement has converged, it must be $V_n^{\pi^*,s}(a) \geq V_n^{\pi,s}(a)$ for any diffusion policy $p^\pi$, $(s, a) \in \mathcal{S} \times \mathcal{A}$ and $n \in \{0, 1, \ldots, N\}$. This completes the proof. □

## D. Continuous-Time Perspective of the Pathwise KL

Song et al. (2021) build the connection between diffusion models and reverse stochastic differential equations (SDEs) that gradually remove noise from data and transform a prior noise distribution into the target distribution. Based on the SDE understanding of diffusion models, we aim to investigate whether the introduced pathwise KL can be extended to continuous-time diffusion modeling.

We define the forward diffusion SDE, which perturbs data into a prior distribution, as the following:

$$
\mathrm{d}x = f(x, t)\mathrm{d}t + g(t)\mathrm{d}w, \tag{39}
$$

where $w$ is the standard Wiener process, $f$ is the drift coefficient, and $g$ is the diffusion coefficient. The reverse process of this SDE is also a diffusion SDE:

$$
\mathrm{d}x = \left[ f(x, t) - g(t)^2 \nabla_x \log p_t(x) \right] \mathrm{d}t + g(t)\mathrm{d}\bar{w}, \tag{40}
$$

where $\bar{w}$ is the reverse-time standard Wiener process and $\mathrm{d}t$ is a infinitesimal *negative* time interval. In practice, we can use a parameterized neural network $s^\theta(x, t)$ to estimate the score of the marginal distribution using score matching:

$$
\theta^* = \arg\min_\theta \mathbb{E}_{t,x(0),x(t)|x(0)} \left[ \lambda(t) \| s_\theta(x(t), t) - \nabla_{x(t)} \log q_{t|0}(x(t)|x(0)) \|^2 \right], \tag{41}
$$

where $q_{t|0}$ is the distribution of $x(t)$ given the initial sample $x(0)$. After the training is completed, we can substitute the score function in Eq. (40) and solve the reverse SDE to obtain samples that follow the target distribution.

We consider two forward SDEs with the same drift and diffusion coefficient, but with different initial distributions $q_0^\pi$ and $q_0^\nu$. Let the approximated score networks be $s^\pi$ and $s^\nu$, and therefore we consider the following reverse SDEs:

1. $dx = \left[ f(x,t) - g(t)^2 s^\pi(x,t) \right] dt + g(t) d\bar{w}$ under measure $\mathbb{P}^\pi$,

2. $dx = \left[ f(x,t) - g(t)^2 s^\nu(x,t) \right] dt + g(t) d\bar{w}$ under measure $\mathbb{P}^\nu$.

Our goal is to compute the KL divergence between the distributions of the two reverse SDEs. Using Girsanov's theorem (Oksendal, 2013), we have

$$
\begin{aligned}
D_{\mathrm{KL}}\left[\mathbb{P}^\pi \| \mathbb{P}^\nu\right] &= \mathbb{E}_{\mathbb{P}^\pi}\left[\log \frac{d\mathbb{P}^\pi}{d\mathbb{P}^\nu}\right] \\
&= \mathbb{E}_{\mathbb{P}^\pi}\left[\int_0^T \frac{\left[\left[f(x,t) - g(t)^2 s^\pi(x,t)\right] - \left[f(x,t) - g(t)^2 s^\nu(x,t)\right]\right]^2}{2g(t)^2} dt\right] \quad \text{(Girsanov's theorem)} \\
&= \mathbb{E}_{\mathbb{P}^\pi}\left[\int_0^T \frac{g(t)^4 \|s^\pi(x,t) - s^\nu(x,t)\|^2}{2g(t)^2} dt\right] \\
&= \mathbb{E}_{\mathbb{P}^\pi}\left[\int_0^T \frac{g(t)^2}{2}\|s^\pi(x,t) - s^\nu(x,t)\|^2 dt\right].
\end{aligned}
\tag{42}
$$

Instantiating the SDE with Variance Preserving noise schedule (Song et al., 2021):

$$
\begin{aligned}
f(x,t) &= \frac{1}{2}\beta(t)x, \\
g(t) &= \sqrt{\beta(t)},
\end{aligned}
\tag{43}
$$

we have

$$
D_{\mathrm{KL}}\left[\mathbb{P}^\pi \| \mathbb{P}^\nu\right] = \mathbb{E}_{\mathbb{P}^\pi}\left[\int_0^T \frac{\beta(t)}{2}\|s^\pi(x,t) - s^\nu(x,t)\|^2 dt\right].
\tag{44}
$$

Eq. (42) presents the analytical form of the KL divergence between two reverse diffusion processes. In the following content, we continue to demonstrate that the pathwise KL introduced in this paper corresponds to Eq. (42) in the limit of $N \to \infty$.

Examining the pathwise KL objective defined in Eq. (11),

$$
\begin{aligned}
&D_{\mathrm{KL}}\left[p_{0:N}^\pi \| p_{0:N}^\nu\right] \\
&= \mathbb{E}_{p_{0:N}^\pi}\left[\sum_{n=1}^N D_{\mathrm{KL}}\left[p_{n-1|n}^{\pi, x^n} \| p_{n-1|n}^{\nu, x^n}\right]\right] \\
&= \mathbb{E}_{p_{0:N}^\pi}\left[\sum_{n=1}^N \frac{\|\mu^\pi(x^n, n) - \mu^\nu(x^n, n)\|^2}{2\sigma_n^2}\right] \\
&= \mathbb{E}_{p_{0:N}^\pi}\left[\sum_{n=1}^N \frac{\beta_n^2}{2\sigma_n^2(1 - \beta_n)}\|s_n^\pi(x^n) - s_n^\nu(x^n)\|^2\right] \\
&= \mathbb{E}_{p_{0:N}^\pi}\left[\sum_{n=1}^N \frac{\beta_n(1 - \bar{\alpha}_n)}{2(1 - \bar{\alpha}_{n-1})(1 - \beta_n)}\|s_n^\pi(x^n) - s_n^\nu(x^n)\|^2\right],
\end{aligned}
\tag{45}
$$

where $s_n$ is the score function at step $n$.

Define $\Delta t = \frac{T}{N}$, by Song et al. (2021), we can show that in the limit of $N \to \infty$, we have $\beta_n = \beta(\frac{nT}{N})\Delta t \to 0$,

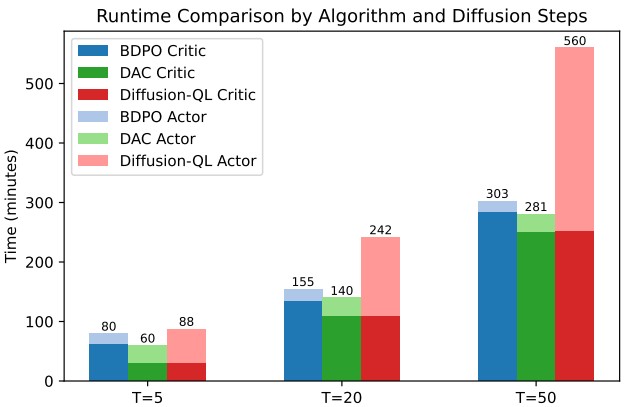

*Figure 11.* Algorithm Runtime of `BDPO`, DAC and Diffusion-QL.

$x^n = x(\frac{nT}{N})$, and $s_n(x^n) = s(x(\frac{nT}{N}), \frac{nT}{N})$. Therefore,

$$
\begin{aligned}
\lim_{N \to \infty} D_{\mathrm{KL}} \left[ p_{0:N}^\pi \| p_{0:N}^\nu \right] &= \lim_{N \to \infty} \mathbb{E}_{p_{0:N}^\pi} \left[ \sum_{n=1}^N \frac{\beta_n(1 - \bar{\alpha}_n)}{2(1 - \bar{\alpha}_{n-1})(1 - \beta_n)} \| s_n^\pi(x^n) - s_n^\nu(x^n) \|^2 \right] \\
&= \lim_{N \to \infty} \mathbb{E}_{p_{0:N}^\pi} \left[ \sum_{n=1}^N \frac{\beta_n}{2} \| s_n^\pi(x^n) - s_n^\nu(x^n) \|^2 \right] \\
&= \lim_{N \to \infty} \mathbb{E}_{p_{0:N}^\pi} \left[ \sum_{n=1}^N \frac{\beta(\frac{nT}{N})}{2} \| s^\pi(x(\frac{nT}{N}), \frac{nT}{N}) - s^\nu(x(\frac{nT}{N}), \frac{nT}{N}) \|^2 \Delta t \right] \\
&= \mathbb{E}_{p^\pi} \left[ \int_0^T \frac{\beta(t)}{2} \| s^\pi(x(t), t) - s^\nu(x(t), t) \|^2 \mathrm{d}t \right],
\end{aligned}
\tag{46}
$$

which exactly corresponds to the KL divergence of SDE in Eq. (42).

## E. Supplementary Experiment Results

### E.1. Discussions about Training Time

A systematic breakdown and comparison of the runtime between `BDPO`, DAC, and Diffusion-QL is illustrated in Figure 11. We evaluate `BDPO`, DAC, and Diffusion-QL with workstations equipped with NVIDIA RTX 4090 cards and the *walker2d-medium-replay-v2* dataset.

Our method consists of three distinct subroutines. The first one is pre-training the behavior diffusion with standard diffusion loss. Our observation is that this phase accounts for a minor fraction (about 8 minutes) of the overall runtime, so we ignore it in the following discussion. The second one is training value functions, including $Q^\pi$ and $V^\pi$. The approach to training $Q^\pi$ is identical to established methods such as Diffusion-QL (Wang et al., 2023) and DAC (Fang et al., 2024), which is essentially sampling diffusion paths at the next state $s'$ and calculating the temporal difference target. In addition to $Q^\pi$, our method additionally trains $V^\pi$. However, this supplementary computation is not resource-intensive, as it only requires single-step diffusion to compute the training target $V^\pi$. Therefore, the additional cost of training diffusion value functions $V^\pi$ is a constant that does not scale with diffusion steps. The third subroutine is training the actor. Akin to EDP (Kang et al., 2023), `BDPO` only requires a single-step diffusion rollout, and thus the cost of training the actor is also a constant that does not scale with diffusion steps. In contrast, Diffusion-QL needs to differentiate with respect to the whole diffusion path, which causes a drastic increase in actor runtime.

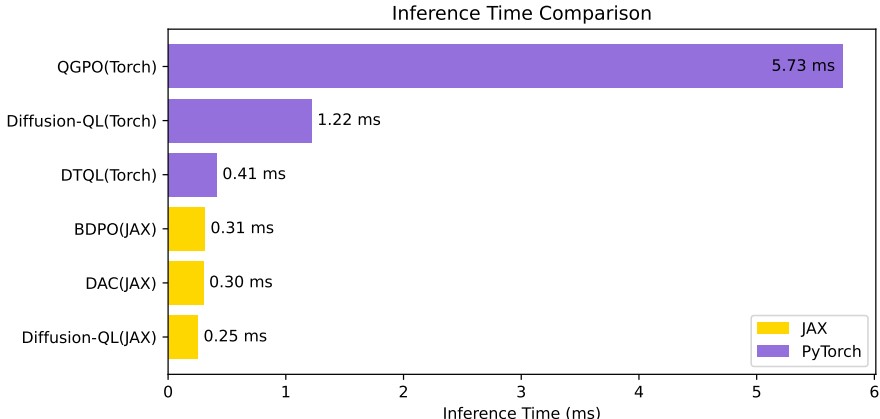

*Figure 12.* Inference Time Comparison.

## E.2. Discussion about Inference Time

Apart from training time, we also evaluate inference time since the latency is critical in real-world applications like robotic control. To benchmark the inference time accurately, we sampled 10,000 states from the *walker2d-medium-replay-v2* dataset, executed the diffusion policies with a batch size of 1, and calculated the average inference time across 10,000 trials. Figure 12 compares the inference speed of `BDPO` and baseline algorithms. Since the specific choice of deep learning backend may have a significant impact on computational latency, we explicitly indicate the backend used for each result.

Among diffusion-based methods, Diffusion-QL generates actions by traversing the reverse diffusion process, and `BDPO` and DAC generate $N = 10$ actions in parallel, followed by Q-value-based sample selection. Given that GPU acceleration effectively amortizes the cost of parallel generation and the calculation of $Q$ values only requires inference through a relatively small network, `BDPO` and DAC are about 10% slower than JAX-based Diffusion-QL. In the meantime, implementations based on PyTorch are much slower than JAX-based implementations due to the dynamic graph execution. The computational overhead is most pronounced in QGPO, which incurs substantial inference latency due to its classifier-guided generation process.

## E.3. Generation Path On Synthetic 2D Tasks

We plot the generation results of `BDPO` on all synthetic 2D datasets in Figure 13. Throughout these datasets, we witness a close resemblance between the final sample distribution and the ground truth target distribution as depicted in Figure 9. The sampling begins with gradual movements across the 2D plane and converges in later diffusion steps. This pattern is further demonstrated by the diffusion value functions, which offer weaker guidance during the initial steps but provide stronger guidance in the final steps. On the contrary, the generation results (Figure 14) of DAC, which leverages $\nabla_{a^n} Q^\pi(s, a^n)$ rather than the gradient of diffusion values $\nabla_{a^n} V_n^{\pi,s}(a^n)$ to guide the generation, fails to capture the target distribution. This is consistent with the findings from QGPO (Lu et al., 2023), which demonstrates that the guidance in `BDPO` is *exact* energy guidance, while the guidance used by DAC is not.

## E.4. Training Curves of D4RL Datasets

The curves of evaluation scores on D4RL datasets are presented in Figure 15 and Figure 16.

## E.5. Ablation on Policy Parameterization

Full results of different policy parameterizations are presented in Figure 17, from which we found that the diffusion policy achieves the best overall performance. Note that we kept the hyperparameter tuning effort the same across different methods (see Section B.3).

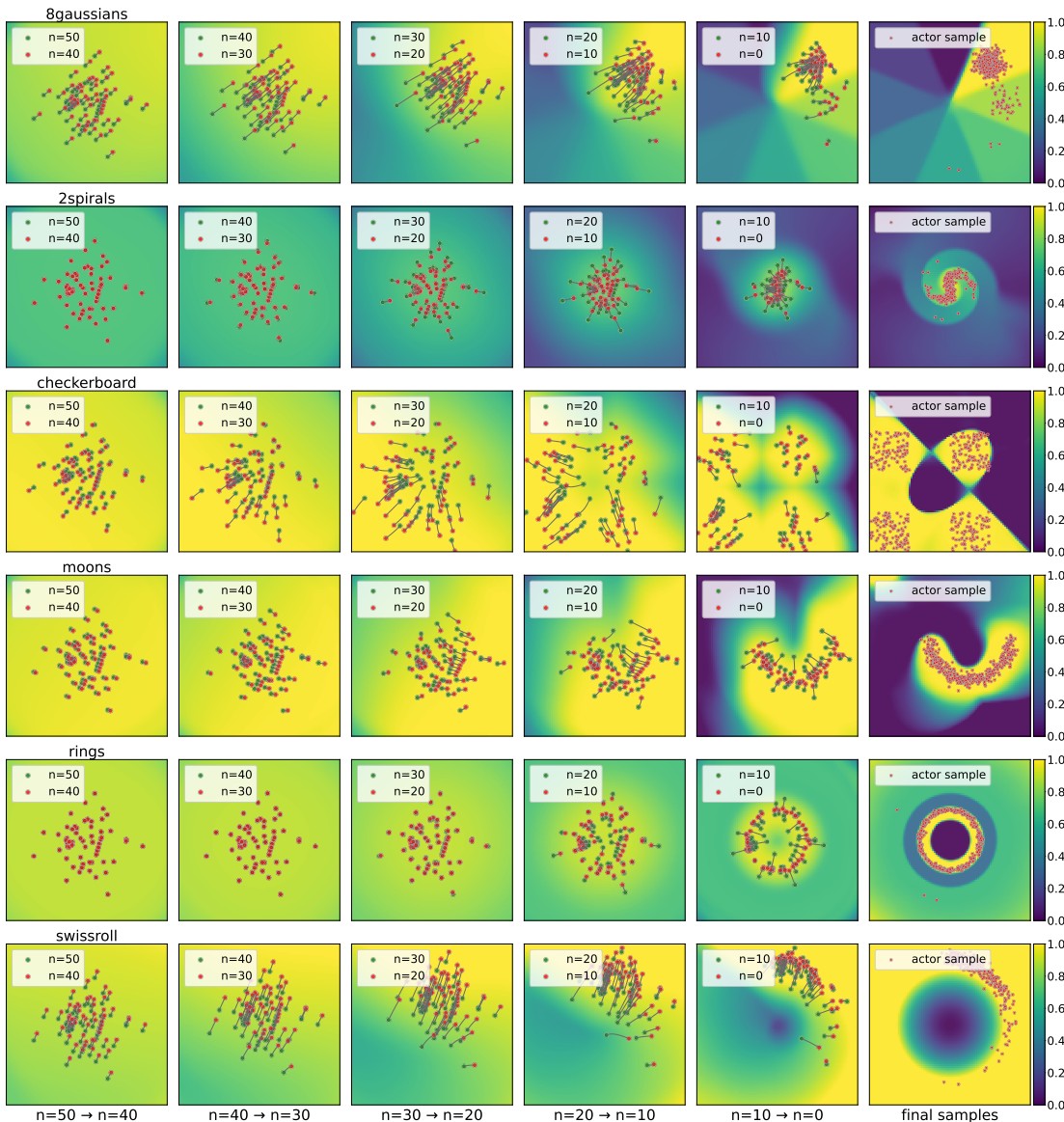

*Figure 13.* Illustration of the diffusion policy and the diffusion value function from `BDPO` on synthetic 2D datasets. The regularization strength is set to $\eta = 0.06$.

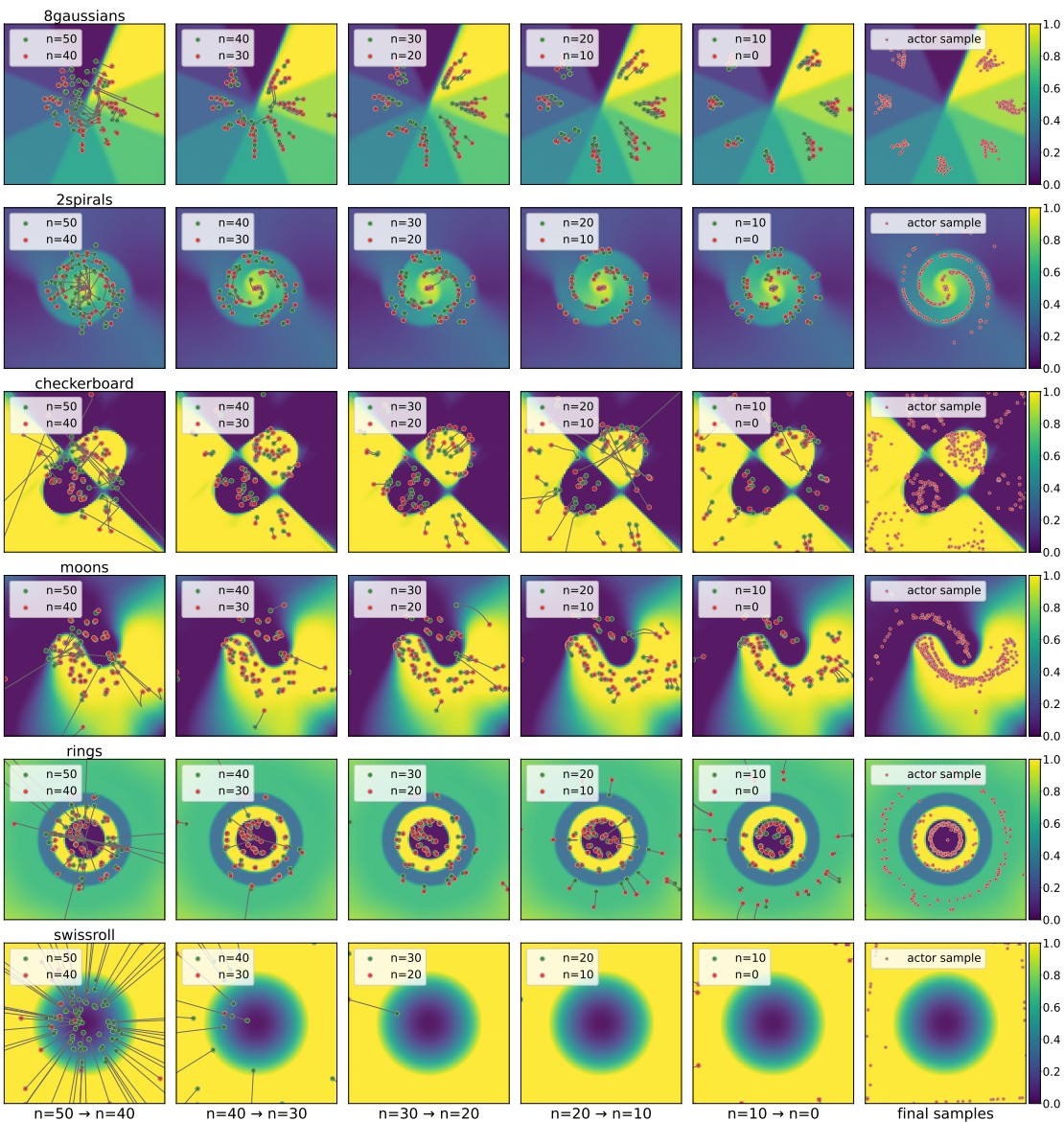

*Figure 14.* Illustration of the diffusion policy and the Q-value function from DAC on synthetic 2D datasets. The regularization strength is set to $\eta = 0.06$.

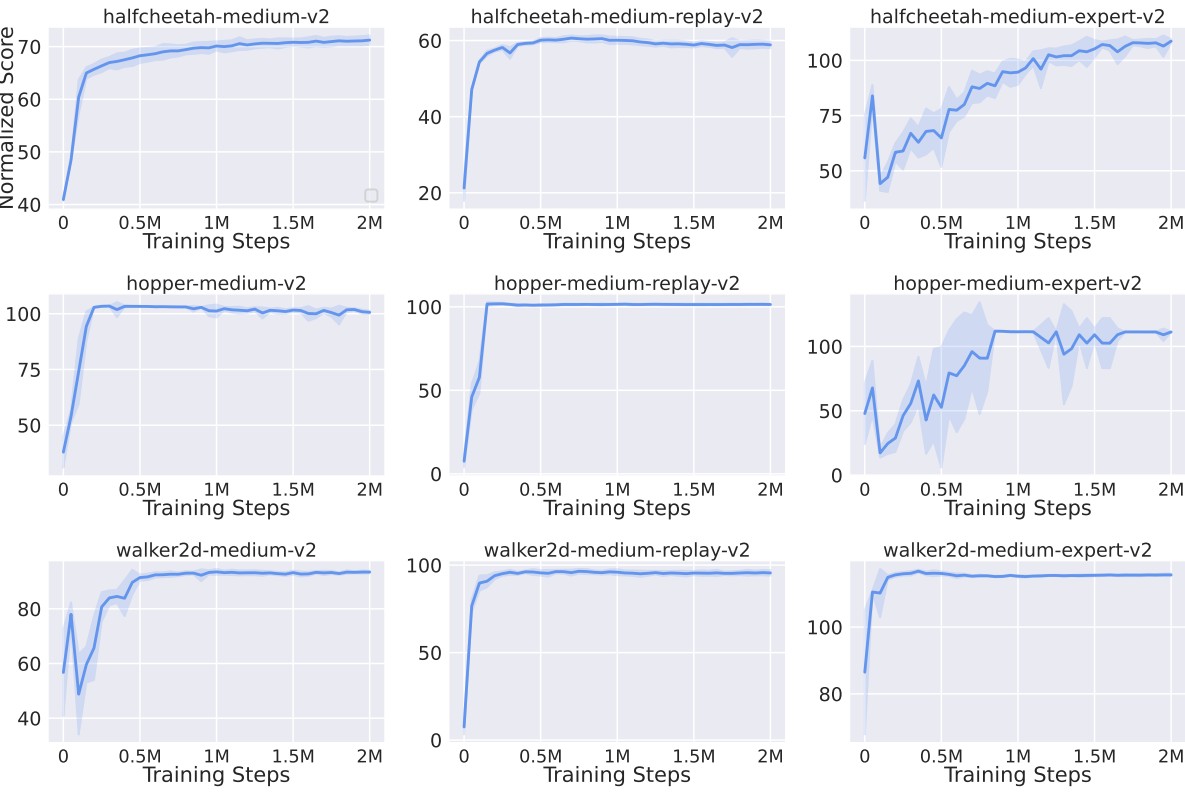

*Figure 15.* Evaluation scores of `BDPO` on D4RL Locomotion datasets. Results are aggregated using 5 independent seeds and 10 evaluation episodes for each seed.

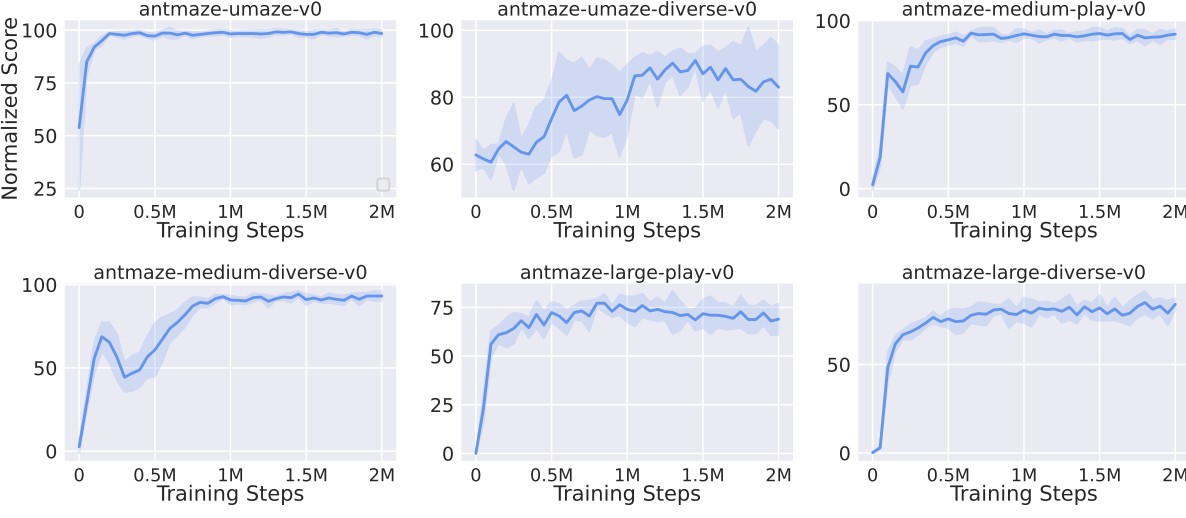

*Figure 16.* Evaluation scores of `BDPO` on D4RL Locomotion datasets. Results are aggregated using 5 independent seeds and 100 evaluation episodes for each seed.

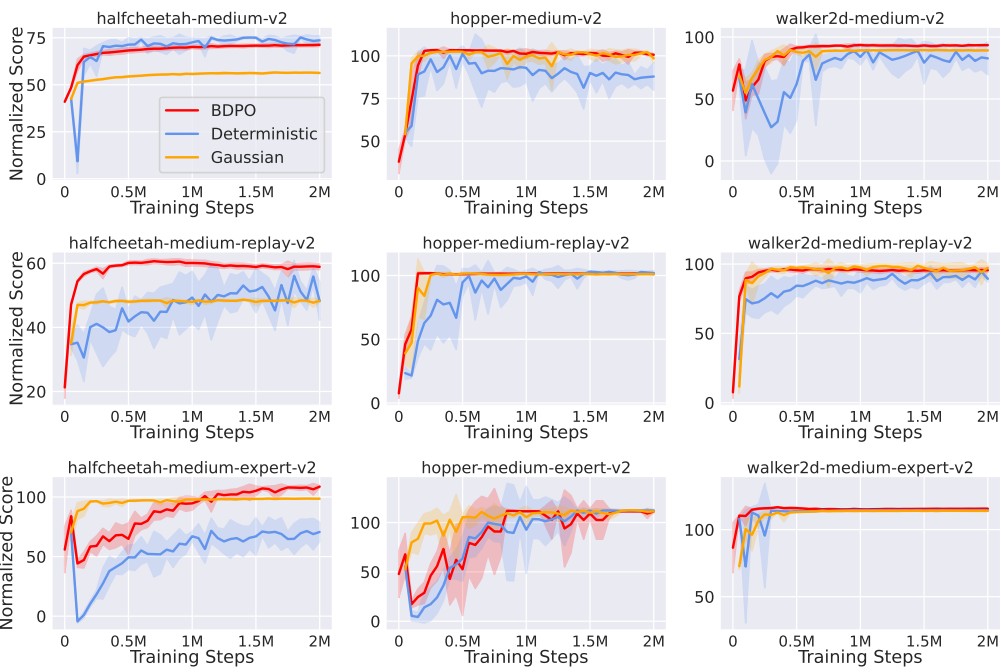

Figure 17. Ablation study with the policy parameterizations. Results are aggregated using 5 independent seeds and 10 evaluation episodes for each seed.

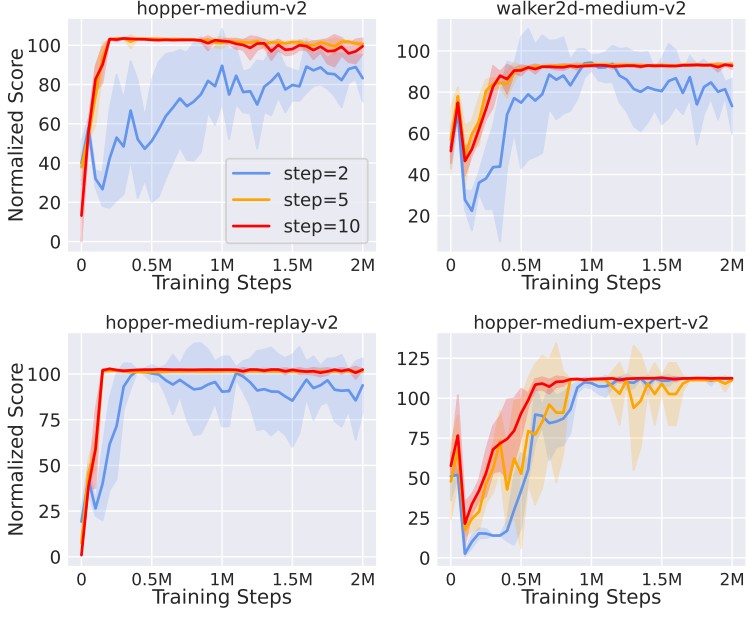

Figure 18. Ablation study with the number of diffusion steps $N$. Results are aggregated using 5 independent seeds and 10 evaluation episodes for each seed.

### E.6. Ablation on Diffusion Steps $N$

The number of diffusion steps $N$ impacts both generation quality and divergence calculation. We empirically evaluate three configurations ($N = 2, 5$, and $10$) in Figure 18. Generally, the performance of `BDPO` degrades substantially when $N$ is too small; however, the improvement diminishes when $N$ exceeds a certain threshold. Based on these observations, we adopt $N = 5$ as our default choice to trade off the efficiency and performance.

