# OpenReview forum: "Behavior-Regularized Diffusion Policy Optimization for Offline Reinforcement Learning"
_ICML.cc/2025/Conference — ICML 2025 poster_

### Official Review · Reviewer_jbHb · 2025-02-26

**Overall Recommendation:** 4

**Summary:**

This paper proposes a diffusion model optimization method based on multi-diffusion-step regularization, which is different from previous behavior-regularized policy methods.

**Claims And Evidence:**

Most of the claims in this paper are well supported by theory and experiments, but some parts remain difficult to understand. Please refer to the questions and comments.

**Essential References Not Discussed:**

None

**Experimental Designs Or Analyses:**

Please refer to the questions and comments for experimental concerns.

**Methods And Evaluation Criteria:**

This paper compares several classic offline RL methods. The choice of experimental environments is appropriate.

**Other Comments Or Suggestions:**

1. In line 817: Regarding the generation process as a decision sequence and the final step reward is Q(s, a^0), then the optimal value function V^{\*,s}_0=Q(s,a^0), Q^{\*,s}_0=Q(s,a^0)$, and $V^{\*,s}_{n}(a^n)=\eta*\log \mathbb{E}_{a^{n-1}}[exp(Q^{\*,s}_n(a^n)/\eta)], Q^{*,s}_n(a^n)=0+1.0*\mathbb{E}_{a^{n-1}}[ V^{*,s}_{n-1} (a^{n-1})]. (1) The formula of V^{*,s}_n(a^n) is different from the formula of line 820. (2) From the recursion, there exists expectation of a^{n-1}, a^{n-2},…, but I can not find the expectation in Equation (26). I suggest the authors show details of the derivation here to help readers understand these important results.
2. In Algorithm 1: If understanding right, when updating the related networks, the training process contains the following steps: 1) Calculate the value Q(s, a^0) and use the results as target to train value V(s, a^0); 2) Get perturbed data a^n by performing forward diffusion process; 3) Run diffusion once to obtain a^{n-1} and use a^n and a^{n-1} to update V function; 4) Calculate the Q target: R+\gamma*V(s, a^n) and update the Q function parameters. If the above process is right, I suggest the author revise Algorithm 1 to make the training process much clearer.

**Other Strengths And Weaknesses:**

Please refer to the questions and comments for review concerns.

**Questions For Authors:**

1. To better understand the background of diffusion models and correspond to the contents mentioned in line 142 right column, I suggest to write Equation (7) as q_{n-1|n}(x^{n-1}|x^n,x^0) rather than q_{n-1|n}(x^{n-1}|x^n).

**Relation To Broader Scientific Literature:**

Previous studies on behavior-regularized policy optimization often apply end-to-end regularization directly on the policy, i.e., focusing on the output actions. In contrast, for expressive diffusion models, this paper considers applying regularization at each generation step of the diffusion model, achieving better-constrained policy optimization.

**Theoretical Claims:**

I have read the theoretical parts related to the paper, including both the main body and the appendix. However, some questions still remain. I have summarized them in the questions and comments.

---

> ### Author Rebuttal · Authors · 2025-04-01
>
> We sincerely appreciate your insightful feedback, which has greatly improved our work. We hope the following clarifications can further address your concerns and enhance your evaluation of our paper.
>
> **Q1: About the proof in Theorem C.1**
> First, we would like to restate the question to address any misunderstandings. Let $Q(s, \cdot)$ denote the ending reward in the diffusion MDP with a discount factor of 1. The reviewer considers the following relationships between the value functions:
> $$\begin{aligned}
> V^{\star, s}\_0&=Q(s, a^0)\\\\
> Q^{\star, s}\_0&=Q(s, a^0)\\\\
> V^{\star, s}\_n(a^n) &= \eta\log\mathbb{E}\_{a^{n-1}}\left[\exp(Q^{\star, s}\_n(a^n)/\eta)\right]\\\\
> Q^{\star, s}\_n(a^n)&=0 + 1.0\mathbb{E}\_{a^{n-1}}\left[V^{*, s}\_{n-1}(a^{n-1})\right]
> \end{aligned}$$
> which seems to contradict our results in line 820. However, we believe the main confusion here is that in the diffusion MDP, the “state” is actually the tuple $(s, a^{n})$ and the state transition is implicit. The policy follows the Gaussian distribution $p^{\pi,s}(a^{n-1}|a^n)$, and upon selecting action $a^{n-1}$, the state instantly transitions to $(s, a^{n-1})$ deterministically. Therefore, the $Q$ function of the diffusion MDP should be defined w.r.t the “state” $(s, a^n)$ and action $a^{n-1}$, leading to the corrected relationships:
> $$\begin{aligned}
> V^{\star, s}\_0(a^0)&=Q(s, a^0)\\\\
> V^{\star, s}\_n(a^n) &= \eta\log\mathbb{E}\_{a^{n-1}}\left[\exp(Q^{\star, s}\_n(a^n, a^{n-1})/\eta)\right]\\\\
> Q^{\star, s}\_{n}(a^n, a^{n-1})&=0 + 1.0V^{\star, s}\_{n-1}(a^{n-1})
> \end{aligned}$$
> Substituting the last equation into the second leads to the formulation in line 820
> $$V^{\star, s}\_n(a^n) = \eta\log\mathbb{E}\_{a^{n-1}}\left[\exp(V^{\star, s}\_{n-1}(a^{n-1})/\eta)\right]。$$
> As for question (2), expanding the recursion gives
> $$\begin{aligned}
> V^{\star, s}\_n(a^n) &= \eta\log\mathbb{E}\_{a^{n-1}\sim p^{\nu,s}\_{n-1|n}}\left[\exp(V^{\star, s}_{n-1}(a^{n-1})/\eta)\right]\\\\
> &= \eta\log\mathbb{E}\_{a^{n-1}\sim p^{\nu,s}\_{n-1|n},a^{n-2}\sim p^{\nu,s}\_{n-2|n-1}}\left[\exp(V^{\star, s}\_{n-2}(a^{n-2})/\eta)\right]\\\\
> &=\ldots\\\\
> &=\eta\log\mathbb{E}\_{a^{n-1}\sim p^{\nu,s}\_{n-1|n}, \ldots, a^0\sim p^{\nu,s}\_{0|1}}\left[\exp(V^{\star, s}\_{0}(a^{0})/\eta)\right]\\\\
> &=\eta\log\int p^{\nu,s}\_{0,1,\ldots,n-1|n}(a^0, a^1, \ldots, a^{n-1}|a^n)\exp(V^{\star, s}\_0(a^0))\mathrm{d} a^{n-1}a^{n-2}\ldots a^{0}\\\\
> &=\eta\log\int p^{\nu,s}\_{0|n}(a^0|a^n)\exp(V^{\star, s}\_0(a^0))\mathrm{d}a^{0}\\\\
> &=\eta\log\mathbb{E}\_{a^{0}\sim p^{\nu,s}\_{0|n}}\left[\exp(V^{\star, s}\_{0}(a^0)/\eta)\right]
> \end{aligned}$$
> where the last but one equation is due to the marginalization over intermediate actions $a^1, a^2, \ldots, a^{n-1}$.  We acknowledge that the derivation in the appendix is somewhat vague and will revise it to explicitly incorporate a proof akin to MaxEnt RL theory for clarity.
>
> **Q2: About the update of value networks.**
> The detailed update procedure of the value networks consists of two steps:
> 1 ) The first step is updating $Q^{\pi}$. To calculate the target, we use the actor diffusion policy to generate paths $a’^{0:N}$ at the next state $s’$, calculate the target value $Q(s’, a’)$ and accumulated penalties along the path $\sum_{n=1}^N\ell^{\pi,s'}_{n}(a^n)$, and perform temporal difference update as per Eq. 12;
> 2 ) The second step is updating $V^{\pi,s}$. This is achieved by sampling $n$ and $a^n$ using forward process, performing one-step diffusion to obtain $a^{n-1}$, and update according to Eq. 14. For $n=1$, we directly use $Q(s, a^0)$ as the target, rather than additionally regressing the output of $V^{s}_0(\cdot)$ to $Q(s, \cdot)$.
>
> Crucially, the update of $Q^{\pi}$ does not depend on $V^{\pi,s}_N$. The diagram of the update is presented in Figure 3. Compared to other methods, the additional cost w.r.t. value function training comes from the second step, which is a constant that does not scale with diffusion steps (see Figure 10).
>
> **Q3: Eq. 7 should be $q\_{n-1|n}(x^{n-1}|x^n,x^0)$**
> We appreciate the suggestion and would like to note that the actual distribution we wish to approximate is the posterior without conditioning on $a^0$, as $a^0$ is unknown during generation. When $a^0$ is given, the posterior distribution $q\_{n-1|n, 0}$ is tractable, with its analytical form being:
> $$q\_{n|n-1, 0}(a^n|a^{n-1}, a^0)=\mathcal{N}(a^{t-1}; \frac{\sqrt{\bar{\alpha}\_{n-1}}\beta_n}{1-\bar{\alpha}\_n}a^0+\frac{\sqrt{\alpha\_n}(1-\bar{\alpha}\_{n-1})}{(1-\bar{\alpha}\_n)}a^n, \sigma\_n I).$$
> Since exact $q\_{n-1|n}$ is intractable, we turn to a parameterized distribution $p^\theta\_{n-1|n}$ and optimize it towards $q\_{n-1|n, 0}$ via in Eq. 10. As shown in DDPM, this training objective yields $p^\theta\_{n-1|n} \approx q\_{n|n-1}$. We recognize that the current presentation lacks clarity and will revise the text to explicitly articulate the connection between $q\_{n-1|n, 0}$ and $q\_{n-1|n, 0}$ in the next version.

---

> > ### Comment · Reviewer_jbHb · 2025-04-04
> >
> > I thank the authors for their explanations. Most of my concerns are addressed. Based on the response, I have several questions:
> >
> > 1. When updating Q, you have sampled the action path $a^{0:N}$ and obtained the KL divergence of $l_n(a^n)$, why don’t you use $\sum_n:0 l(a^n)+Q(a^0)$ directly as the V target of $V(a^n)$?
> > 2. The training of V is bootstrapped (Equation (14)). Under random sampling of $a^n$, could this training approach lead to instability in V, since the $a^n$ is unlikely to be sampled?
> > 3. Have the authors considered treating the generation process as a low-level MDP and the RL MDP as a high-level MDP? In this case, the divergence in the generation process could be viewed as the reward of the low-level MDP, resulting in a reward sequence such as $...,r(s,a),-l_N,-l_{N-1},...,-l_1,r(s',a'),...$. And directly training Q and $\pi$ on this type of reward sequence.

---

> > > ### Author Response · Authors · 2025-04-05
> > >
> > > We are glad to know that most of the concerns have been addressed. Below, we provide detailed answers to the remaining questions, and we hope these can further support your evaluation of our work.
> > >
> > > 1. Why don't you reuse the already sampled diffusion path and use $Q(a^0)-\eta\sum_{i=1}^{n}l^{\pi,s}(a^n)$ to update $V^{\pi,s}(a^n)$?
> > > We sincerely appreciate your suggestion. We acknowledge that taking $Q(a^0)-\eta\sum\_{i=1}^{n}l^{\pi,s}(a^n)$ as the target is also a theoretically valid solution. The rationale behind us using a separate sampling process to compute the target for $V^{\pi,s}\_n(a^n)$ is two fold: 1) Using single-step diffusion generation and bootstrapping from $V^{\pi,s}\_{n-1}(a^{n-1})$ is more consistent with how we update the diffusion policy; 2) Our approach supports multi-sample estimation of the target. Specifically, by sampling multiple $a^{n-1}$ and averaging their values $V\_{n-1}(a^{n-1})$, we achieve a more accurate approximation of the expected value in Eq. 14.  In practice, we sampled N=10 actions $a^{n-1}$ for each $a^n$. However, re-using the diffusion path only gives single sample estimation of the target value for each $a^n$, which may lead to higher variance.
> > > We present the learning curves (averaged over 3 random seeds) of our update scheme (labeled 'Ours') and the proposed alternative (labeled 'New') in this link: https://f005.backblazeb2.com/file/bdpo-review/rebuttal_abla_valueupdate.pdf , which reveals that our method achieves slightly faster convergence and stability in performance.
> > >
> > > 2. Since $a^n$ are randomly sampled, will the training approach cause instability in $V$?
> > > We plot the curves of output values of $V$ in the following anonymous link: https://f005.backblazeb2.com/file/bdpo-review/rebuttal_v_values.pdf . Please note that the value network output may fluctuate somewhat in the first 50k steps, because we begin training the policy after 50k steps. Overall we found that the value network is very stable. We attribute this stability to three key factors: 1) as discussed earlier, we sample N=10 actions and take their average value as the target, which produces estimations with lower variances. 2) the value function $V$ is optimized on *noisy actions* $a^n$, which are generated by perturbing dataset actions $a^0$ with Gaussian distributions and therefore have infinite support over the action space. In this sense, $V$ is optimized on intermediate actions from all over the action space. 3) As evidenced in Figure 4 and Figure 11, the value function exhibits much smoother outputs over the action space at higher noise levels, which inherently reduces variance in value function updates.
> > >
> > > 3. Have the authors considered treating the generation process and the RL MDP as the low-level and high-level MDP, respectively?
> > > Yes, we have explored this approach and trained a unified value function $Q(s, a, n)$ on the extended reward sequence. The most intriguing property of this formulation is that the value function udpate no longer requires sampling the full diffusion path -- instead, it only needs one single diffusion step, thereby reducing the computational overhead of critic learning to a constant that does not scale with the number of diffusion steps.
> > > However, in preliminary experiments, we observed that this formulation consistently underperforms our method. The primary reason is that unfolding the generation MDP into the RL MDP essentially extends the horizon by a factor of N (number of diffusion steps), making TD learning over such an extended horizon significantly more challenging. In contrast, our approach maintains the original horizon of the environment MDP for $Q$-function, while treating the diffusion MDP as "branches" from the environment MDP. This design ensures more stable and better performance (curves are presented in Figure 12). Alternatively, our method can be interpreted as employing an N-step return as the target of $Q$. Specifically, we compute the cumulative sum of rewards and penalties over N steps of the diffusion MDP and then bootstrap from the subsequent environment state.

---

### Official Review · Reviewer_iYRy · 2025-03-12

**Overall Recommendation:** 3

**Summary:**

This paper use diffusion policis for offline RL and the main idea is to take the rollout quality of the diffusion process as extra regularization. In other words, for behavior cloning, the paper proposes to measure the similarity between the demonstration action and the learnt action by comparing their distributions of trajectories generated by the revserse procedure of the diffusion process. They show that such difference can be accumulated as another scale of value function and is useful to behavior cloning.

**Claims And Evidence:**

1. To justify the proposed method, the paper says that optimizing eq.11 is equivalent to eq.1, which means that the behavior of the proposed method would be like a behavior clone method which takes eq.1 as its objective. However,  what we really interested in is the optimality condition under the offline RL setting - in other words, can an optimal policy be delivered by optimizing eq.11?

2. The proposed method provides a stronger regularization for the divergence between the new policy and the behavior policy with the pathwise constraint. This would make the policy more likely to be suffered from the bad behaviors generated by the sub-optimal behavior policy. Could the author discuss more about this point?

3. the authors consider the value of the actions generated during the diffusion procedure, however, such actions are artifical and are never been actually taken in practice, so how can you ensure that they would be appeared in the offline dataset? (assumption 4.3) In this sense, lots of OOD action would be evaluated by the proposed method, how to ensure the algorithm's stablity?

**Essential References Not Discussed:**

1. In the literature of deep learning on teacher student network, there are several works about recording the learning trajectory of the tearcher network, and asking the student network to imitate that learning behavior. This paper obviously follows the same idea - the new policy should imitate the diffusion trajectory of the behavior policy. So I think the paper lacks the reference on such works.

**Experimental Designs Or Analyses:**

In Table 1, there lack the results on 'random' and 'expert' benchmarks, which is important as well.

**Methods And Evaluation Criteria:**

yes

**Other Comments Or Suggestions:**

n/a

**Other Strengths And Weaknesses:**

n/a

**Questions For Authors:**

see above.

**Relation To Broader Scientific Literature:**

see below

**Theoretical Claims:**

1. Assumption 4.3. This assumption is like the well-known Concentrability assumption, which means that the dataset should cover the visitation states of any policies from the candidate set. When the set is large enough, the coverage of dataset would be very large, which is hard to realize in practice. As a result, such assumption has been aborted in recent works, where only the optimal coverage assumption should be given. However, in our opinion, the Assumption 4.3 is even more stronger than the Concentrability assumption, because it assumes the coverage at each diffusion step. This makes the theoretical results hard to generalize to real-world scenarios.

2. The 'optimal policy' in Theorem 4.2 seems different from that in other works, such as [Bellman-consistent Pessimism for Offline Reinforcement Learning], where the optimal policy is the global optimal policy in the true MDP. Therefore, Theorem 4.2 only guarantees the equivalence of Eq.(1) and (11), in my opinion, which is not so interesting as the analysis of the optimality of the learnt policy.

3. Lack of the theoretical comparison with other works, especially with those works about diffusion-based policy.

---

> ### Author Rebuttal · Authors · 2025-04-01
>
> Thank you for providing valuable feedback. Below, we provide further clarification and results to address your concern and we hope these materials can enhance your evaluation of our paper. **Due to the space constraint, we will post our discussion of the connection between BDPO and broader literatures (Q6 and Q8) in follow-up responses.** You can also access the responses in the anonymous link: https://f005.backblazeb2.com/file/bdpo-review/Q6-andQ8.txt .
>
> **Q1: Can an optimal policy be delivered by optimizing eq.11?**
>
> We would like to clarify that Eq. 1 is also a **well-established RL objective widely adopted in the literature**, and therefore, optimizing Eq. 11 delivers optimal policies with respect to Eq. 1. Please refer to our response to Q5 for details.
>
> **Q2:  The pathwise KL regularization is stronger, making the policy suffer from sub-optimal data?**
>
> Yes, BDPO additionally regularizes intermediate actions along the diffusion path. However, we emphasize that this is **not a stronger regularization**, as we have established the equivalence between pathwise and action-wise constraint in Theorem 4.2. Given this guarantee, the pathwise constraint is in fact preferable as it enables a finer-grained control over the generation process. Furthermore, when the behavior policy is suboptimal, the regularization strength $\eta$ can be decreased to allow greater exploitation from the value function.
>
> **Q3: Intermediate actions are not present in the dataset.**
>
> When training value functions $V^{\pi, s}\_n$ for $n>0$, we optimize them using actions $a^n$ generated by first sampling clean actions $a^0$ from the dataset and then perturbing them using the forward diffusion process $a^n\sim q\_{n|0}(a^0)$. Consequently, **the action support for $V^{\pi,s}\_{n}$ with $n > 0$ is infinite and spans the entire action space** due to the unbounded support of the Gaussian noise distribution. The only risk of OOD evaluation comes from querying $V^{\pi,s}_0(a^0)$ for some $a^0$ generated by the actor. However, this challenge is inherent to all policy iteration methods and can be effectively mitigated by tuning the regularization strength.
>
> **Q4: The assumption about concentrability is stronger?**
>
> We clarify that Assumption 4.3 is made only to ensure the boundedness of the KL divergence in our theory, and it is not the concentrability used in papers like the reviewer mentioned. Similar assumptions are also made in papers that incorporate behavior regularization -- for example, SAC assumes $|\mathcal{A}| < \infty$ to ensure the policy entropy is bounded.
>
> Regarding the concentrability, as discussed in Q3, the marginal distribution $p^{\nu,s}_n$ at $n>0$ has infinite support over the action space due to Gaussian perturbations. Consequently, the concentrability requirement reduces to the base case of $n=0$, meaning BDPO does not impose stronger concentrability assumptions than other methods. Our primary contribution is providing a practical implementation of the behavior-regularization RL for diffusion-based policies, rather than introducing new theoretical bounds.
>
> **Q5: The optimal policy differs from other works.**
>
> The optimal policy differs because we consider the **behavior-regularized RL framework** (Eq. 1), which augments the standard RL objective with KL divergence. This framework helps us to shape the policy and is widely adopted in RL. For example, when $\nu$ is specified as the uniform distribution or the dataset collection policy, the framework becomes MaxEnt RL (used by SAC) or regularized RL (used by offline RL methods like ReBRAC and XQL), respectively. Other offline RL algorithms (e.g., TD3-BC, AWAC) can also be categorized into this framework if we omit the KL divergence term during policy evaluation. This framework is also adopted by applications like RLHF, where KL regularization towards the reference model prevents model collapse. Therefore, we believe it is important to study this framework.
>
> **Q6: Lack of theoretical comparison with other diffusion-based works.**
>
> See follow-up responses or the link.
>
> **Q7: Experiment results on random and expert datasets**
>
> We compare BDPO against several baselines, including a previous model-free SOTA, ReBRAC and the best performing diffusion-based method, DAC. All results are averaged over 4 seeds.
>
> |Dataset|IQL|CQL|ReBRAC|DAC|BDPO|
> |-|-|-|-|-|-|
> |hc-random|19.5|31.1|29.5|28.6|28.6±0.9|
> |hc-expert|95.5|97.3|105.9|103.4|105.5±2.4|
> |hop-random|10.1|5.3|8.1|8.4|15.0±9.7|
> |hop-expert|108.8|106.5|100.1|98.6|110.2±2.3|
> |walk-random|11.3|5.1|18.4|4.1|0.61±0.15|
> |walk-expert|96.9|109.3|112.3|113.5|110.4±0.1|
> |sum|342.1|354.6|374.3|356.6|374.4|
>
> Overall, BDPO matches SOTA performance (ReBRAC) and outperforms DAC, though the margin is narrow. Note that model-free offline RL methods tend to struggle on random datasets due to the inferior data quality.
>
> **Q8: Lack of references about imitation learning**
>
> See follow-up responses or the link.

---

### Official Review · Reviewer_3ubG · 2025-03-16

**Overall Recommendation:** 3

**Summary:**

The paper introduces Behavior-Regularized Diffusion Policy Optimization (BDPO), a framework for offline RL that integrates diffusion-based policies with behavior regularization. The key innovation is formulating KL regularization along the whole diffusion steps instead of on the final result, enabling more effecient computation. The authors propose a two-time-scale actor-critic algorithm that optimizes value functions, which further improves computational efficiency and training stability. Theoretical results establish equivalence between the pathwise KL regularization and the standard KL-regularized RL objective. Experiments on synthetic 2D tasks and D4RL benchmarks demonstrate superior performance compared to baseline methods, particularly in locomotion and antmaze navigation tasks.

## update after rebuttal

The rebuttal has addressed most of my concerns. Given that the paper is strong with relatively minor deficits, I have decided to maintain my original rating of weak accept.

**Claims And Evidence:**

Yes.

**Essential References Not Discussed:**

No.

**Experimental Designs Or Analyses:**

Yes. I have checked all experimental designs and analyses.

**Methods And Evaluation Criteria:**

Yes.

**Other Comments Or Suggestions:**

Here are some possible typos:
- Title of Figure 8: "Illuration" -> "Illustration".
- Title of Figure 7(a): $\beta$ -> $\eta$.
- Eq. (18), $V(s,a)$ -> $V(a^n)$.

**Other Strengths And Weaknesses:**

Strengths
- The method is novel on incorporating behavior regularization on each diffusion step, while maintaining training efficiency by the two-time-scale scheme.
- The theoretical proof is clear and well-written, which links pathwise KL to standard KL regularization and continuous-time SDEs.
- The empirical results across diverse tasks are strong.

Weaknesses: see questions.

**Questions For Authors:**

- The bi-level TD-learning framework is proposed to avoid preserving the computation graph of all diffusion steps like Diffusion-QL. I wonder have you tried the optimization in EDP[1], which improves the efficiency of Diffusion-QL and also avoid perserving the whole computation graph. Is their method inconsistent with you behavior regularization objective?
- As you use an ensemble of Q networks, how do you select the action in evaluation by the highest Q value? Do you use the mean of these Q values? Also, could you please report the inference time of BDPO, and compare it with your main baselines?
- For two key hyperparameters $\beta$ and $\rho$
  - In what range do you search $\eta$ and $\rho$?
  - Figure 7(a) shows that the trend of BDPO is completely different w.r.t. $\eta$ and $\rho$ in Halfcheetah and Walker2d tasks, though they are both medium dataset. Does this stem from some properties of these two environment? Also, is this trend similar to those on datasets other than medium (e.g., medium-replay)?
  - It seems that smaller $\eta$ in your tested range leads to better performance in Halfcheetah-medium. Could you please test $\eta=0$, i.e., no behavior regularization? The same for $\rho=0$, i.e., no LCB penalty.
  - It seems that BDPO is sensitive to $\rho$, as shown in Figure 7(b). Adjusting $\rho$ from $1.0$ to $0.75$ or $1.5$ will lead to significant drop in performance on Walker2d-medium. Could you please explain this phenomenon?
  - Finally, as exaustively tuning hyperparameters are impractical in many real tasks, could you please provide some insight to how to choose hyperparameters, or at least shrink the search range, for unseen tasks by their task properties?

**Relation To Broader Scientific Literature:**

The work builds on two key strands: behavior regularization and diffusion policies.
- Behavior regularization is commonly used in both online and offline RL to provide an expected objective to the policy. Specifically, previous offline RL works usually utilize the KL divergence between the learning policy and the behavior policy. This paper extends behavior regularization to intermidiate diffusion steps, while keeping high computational efficiency through the two-time-scale optimization scheme. This scheme resembles DPPO but incorporates penalty in each diffusion step.
- The diffusion policies model RL policies by diffusion models, instead of traditional Gaussion policies, As diffusion models enjoy better capability on complicated multi-modal distribution, they have surpassed their counterparts using Gaussion policies. This paper proposes a novel regularization and training scheme to further improve the performance of diffusion policies, while keeping the training complexity relatively low.

**Theoretical Claims:**

Yes. I have checked all proofs for theoretical claims.

---

> ### Author Rebuttal · Authors · 2025-04-01
>
> We would like to thank the reviewer for his/her constructive feedback. Below is further clarification for the reviewer's concern.
>
> **Q1: Using EDP for policy optimization?**
>
> Our policy improvement objective is to maximize the expected Q-values while also minimizing the KL divergence:
> $$\max\_{p^{s,\pi}}\ \mathbb{E}\_{a^0\sim p^{s,\pi}\_0}[Q(s, a^0)]-\eta \mathrm{KL}\left[p^{s,\pi}\_{0:N}\|p^{s,\nu}\_{0:N}\right]$$
> Let us ignore the second KL term for now, since EDP implements this constraint using diffusion loss. For the first term, in order to circumvent backpropagating the gradient of Q through the diffusion path, EDP introduces the following action approximation:
> $$\hat{a} = \frac{1}{\sqrt{\bar{\alpha}\_n}}a^n - \frac{\sqrt{1-\bar{\alpha}^n}}{\sqrt{\bar{\alpha}^n}}\epsilon^{\pi,s}(a^n, n)$$
> where $\epsilon^{\pi,s}$ is the output of the score network. Afterwards, they use $\hat{a}$ as the approximation for $\mathbb{E}\_{a^0\sim p^{s,\pi}\_{0|n}}[Q(s, a^0)]$. However, we emphasize that this approximation is **inexact and biased**. The action approximation is essentially the mean of the posterior distribution:
> $$\hat{a}=\mathbb{E}\_{a^0\sim p^{\pi,s}\_{0|n}}\left[a^0\right]$$
> meaning that they are using $Q(s, \mathbb{E}\_{a^0\sim p^{\pi,s}\_{0|n}}\left[a^0\right])$ to approximate $\mathbb{E}\_{a^0\sim p^{s,\pi}\_{0|n}}[Q(s, a^0)]$, which is biased and inconsistent with our theory.
>
> **Q2: How to select the action with the highest Q value? What about the inference time?**
>
> Yes, we use the average of the ensemble Q networks to select the actions. During inference, BDPO first generates $N_a=10$ candidate actions **in parallel**, calculates their $Q$-values and selects the best one. The following table presents the inference latency per state, averaged over 100K trials:
>
> |Algorithm|Inference Time (ms)|
> |:-----:|:-----:|
> |BDPO (JAX)|0.310|
> |DAC (JAX)|0.298|
> |Diffusion-QL (JAX)|0.251|
> |Diffusion-QL (PyTorch)|1.22|
> |DTQL (PyTorch)|0.411|
> |QGPO (Pytorch)|5.73|
>
> We found that the inference cost of BDPO is comparable to DAC, which also generates 10 actions and selects the best one with the highest Q-value. The JAX implementation of Diffusion-QL is faster than BDPO, since it only generates one action and does not query the Q-values. The PyTorch version of Diffusion-QL is much slower. For DTQL, since its policy network is simply a one-step policy, its inference time is comparable to BDPO. Finally, QGPO requires taking the gradient of the Q-value network to calculate the guidance, which results in heavy computation overhead during inference.
>
> **Q3: About hyperparameters**
> - **The range of parameter sweeping for eta and rho?**
>
>   For locomotion tasks, we swept eta and rho among {0.5, 1.0, 1.5, 2.0}. For antmaze tasks, we swept rho among {0.5, 0.8, 1.0} and eta among {1.0, 5.0, 10.0}.
> - **The different trend w.r.t. the hyper-parameters in halfcheetah and walker2d tasks, and what about other datasets like medium-replay?**
>
>   The sensitivity analysis of eta and rho for medium-replay tasks is provided in the following anonymous links: https://f005.backblazeb2.com/file/bdpo-review/rebuttal_abla_eta.pdf and https://f005.backblazeb2.com/file/bdpo-review/rebuttal_abla_rho.pdf . Overall we found that the trend is similar across most of the datasets, that is, excessively large or small $\rho$ and $\eta$ may result in fluctuation or degradation in the performance. The halfcheetah-medium-v2 dataset seems to be an exception in that it is more tolerant to extreme parameter values.
>  - **Setting $\eta=0$ or $\rho=0$?**
>   For results with $\rho=0$, please refer to the above link to ablation w.r.t. rho. For results with $\eta=0$, we additionally provide curves on medium-expert datasets, and the results are provided in the anonymous link: https://f005.backblazeb2.com/file/bdpo-review/rebuttal_eta0.pdf . The lower confidence bound technique is conceptually similar to the commonly adopted double Q-network trick that penalizes the Q-values of OOD actions, and therefore when there is no LCB penalty, the performance drops sharply due to severe over-estimation. Setting $\eta=0$ also results in performance fluctuation and decrease for walker datasets due to insufficient constraint. However, in halfcheetah datasets, $\eta=0$ improves performance, likely because LCB already penalizes out-of-distribution (OOD) actions effectively.
>  - **About the sensitivity to rho and advice on choosing hyper-parameters?**
>   After inspecting the training details, we observed that an excessively small $\rho$ leads to severe over-estimation in Q-values, while an excessively large $\rho$ causes severe underestimation instead, both of which result in performance degradation. We found that a $\rho$ value yielding stable value estimations generally correlates with strong performance. Thus, we recommend to first adjust $\rho$ until stable value estimations are achieved, and then gradually decrease $\eta$ to strike a balance between robustness and performance.

---

### Decision · Program_Chairs · 2025-05-01

**Decision:**

Accept (poster)

**Comment:**

**Summary:**
This paper presents Behavior-Regularized Diffusion Policy Optimization (BDPO), a principled and efficient offline RL framework that integrates diffusion-based policy modeling with pathwise behavior regularization. By introducing KL regularization across intermediate diffusion steps and employing a two-time-scale actor-critic optimization, the method achieves strong theoretical grounding, efficient training, and superior empirical performance across standard benchmarks.

**Pros:**
1. Proposes a novel and elegant formulation of diffusion policy regularization that improves stability and expressiveness.
2. Achieves strong results across a range of D4RL benchmarks, supported by clear ablation studies and insightful analyses.
3. Theoretical contributions, including the equivalence of pathwise KL regularization and standard objectives, are well-articulated and rigorous.

**Cons:**
Some assumptions may be restrictive in practice, and additional implementation details and comparisons with related baselines would further strengthen the work.